

# The Impact of aerosol–ice nuclei-cloud interactions on a Typical Spring Dust-Precipitation Event in China

Jian Zhang[1,2], Chunhong Zhou[2]*, Xiaoyu Shen[2,3], Hong Wang[2,4], Xiaoye Zhang[2,4]

[1] Key Laboratory for Aerosol-Cloud-Precipitation of China Meteorological Administration, Nanjing
University of Information Science & Technology, Nanjing, Jiangsu, China

[2] Institute of Atmospheric Composition and Environmental Meteorology & Key Laboratory of
Atmospheric Chemistry of CMA, Chinese Academy of Meteorological Sciences, Beijing, China

[3] Key Laboratory of Urban Air Particulate Pollution Prevention and Control of Ministry of Ecology and
Environment, College of Environmental Science and Engineering, Nankai University, Tianjin, China

[4] State Key Laboratory of Severe Weather, Chinese Academy of Meteorological Sciences, Beijing,
China

* Corresponding authors.
E-mail addresses: zhouch@cma.gov.cn (Chunhong Zhou)

**Abstract**.

To investigate the impact of ice nuclei (IN) activated by dust aerosols on precipitation over China, this study uses regional Global/Regional Assimilation and Prediction System – China Meteorological Administration Unified Atmospheric Chemistry Environment (GRAPES/CUACE). The original temperature-dependent IN nucleation scheme is improved by incorporating an on-line aerosol–IN nucleation scheme. The INs are fed on-line into the Double-Moment 6-Class (WDM6) cloud microphysics scheme in a typical dust affected precipitation event in East Asia.

The on-line aerosol–IN nucleation scheme modifies the spatial distribution and density of IN. Compared with the systematic underestimation in original WDM6, INs reach $10^3$–$10^4$ L$^{-1}$ with the improved scheme, and cloud ice is reasonably formed between 2 and 6 km in height.

The scheme alters the distribution of cloud hydrometeors, making it closer to observations. Above the freezing level, the ice-phase hydrometeors mixing ratio decreases due to the higher cloud-top temperatures in dusty weather. And the ratio of cloud ice to cloud snow changes from 1:1 to 1:3. Near the freezing level, increased cloud ice converts to cloud water, resulting in its increasing. During the dust-precipitation event, rainwater is decreased due to vapor competition between IN and



cloud condensation nuclei.

The scheme also modulates the precipitation distribution closer to observations. It suppresses precipitation near dust source areas, where accumulated precipitation decreased by about 1.5 mm, while the downstream precipitation increased by about 0.18 mm.

Keywords: aerosol–IN–cloud–precipitation interactions;dust- precipitation event;on-line aerosol–IN nucleation scheme



**1 Introduction**

The formation of cloud ice is one of the key processes in ice-phase precipitation, and ice nuclei (IN) involving aerosols play a crucial role in the development of cloud ice, particularly in mid- to high-latitude areas and in the upper troposphere (Li et al.,

2022; Chen et al., 2023; Knopf and Alpert, 2023). This is because homogeneous nucleation without IN occurs only below −40 °C, which are relatively rare in natural atmospheric environments (Eastwood et al., 2008; Che et al., 2021). In contrast, heterogeneous nucleation involving IN can occur under ice-supersaturated conditions at much higher temperatures. Therefore, heterogeneous nucleation mediated by IN is

the dominant pathway for cloud ice formation.

Aerosols can act as IN, participating in cloud formation, altering cloud microphysical properties and lifetimes, thereby affecting precipitation (Twomey, 1977; Albrecht, 1989; Andreae and Crutzen, 1997; Ramanathan et al., 2002). Among different species, mineral dust is recognized as one of the primary sources of atmospheric IN

(Khain et al., 2000; Nenes et al., 2014; Khain et al.,2015; Tobo et al., 2019). Dust particles have unique surface structures that facilitate the adsorption and binding of water molecules, promoting the formation of cloud ice (Possner et al., 2017; Stevens et al., 2018). Stith et al. (2009) and DeMott et al. (2015) have found a high correlation between IN number concentration and aerosols with diameters larger than 0.5 μm, with

mineral dust accounting for 33-50% of the total IN. Jiang et al. (2016) found that IN concentrations observed during dust events in Huangshan and Nanjing were significantly higher than those during non-dust conditions. Tobo et al. (2020) observed that IN concentrations increased remarkably during dust events in Tokyo when temperatures were above −25 °C. In addition, aged dust aerosol has increased solubility,

which can act as cloud condensation nuclei (CCN) and thereby further influencing precipitation (Trochkine et al., 2003).

Compared with the relatively well-understood impacts of aerosols as CCN, the role of dust as IN is considerably more complex and remains poorly understood, with substantial uncertainties (Kaufman et al., 2002; Pan et al., 2017). From statistical

studies, Han et al. (2008) found that precipitation events often co-occur with dust storm



in the Taklimakan Desert, showing a significant negative correlation between dust storm frequency and precipitation at interannual scales but a positive correlation at monthly scales. Based on observations data from Semi-Arid Climate and Environment Observatory of Lanzhou University, Wang (2013) found that dust aerosols tend to suppress precipitation over arid and semi-arid areas in spring, while promoting it in summer. From Observational studies, satellite and aircraft measurements by Rosenfeld and Bell (2011) found that dust had little impact on total cloud water content but reduces cloud droplet effective radius and precipitation efficiency. Naeger (2018)found that dust could enhance precipitation over Florida through multi-sensor satellite observations and field campaigns. Hu et al. (2023) found that the influence of springtime dust on precipitation was modulated by other aerosols. Overall, due to the multi-factors influencing precipitation beyond aerosols, it remains challenging to quantify the impact of dust on precipitation by observational solely (Zhou et al., 2016; Stier et al., 2024).

Numerical model is a crucial approach for numerically studying the impact of dust on precipitation. In early cloud microphysics scheme, the ice nucleation scheme did not account for aerosols, with IN concentrations typically expressed as functions of temperature or supersaturation (DeMott et al., 2010). Moreover, many clouds ice microphysical schemes were single-moment, which only simulated the mass mixing ratio of cloud ice. This single-moment schemes often led to large biases in cloud ice mass concentrations (Molthan and Colle, 2012). In contrast, double-moment ice schemes that simulate both cloud ice mass and number concentrations provide more accurate cloud ice particle size distributions and concentrations that are more consistent with MODIS satellite observations(Park and Lim, 2023; Kwon et al., 2023). The double-moment ice schemes can provide more stable and improved precipitation simulations (Kang et al., 2018; Shen et al., 2022; Shen et al., 2024). Mascioli et al. (2021) used the Thompson aerosol-aware microphysics scheme, incorporating the IN nucleation scheme of DeMott et al. (2010), to study the sensitivity of precipitation to different prescribed dust aerosol concentrations. Park and Lim (2023) implemented a





cloud ice microphysics scheme in the Weather Research and Forecasting Double-Moment 6-class (WDM6) microphysics scheme and examined the influence of dust on precipitation using aerosol diffusion coefficients. Their results suggested that dust could modulate the spatial distribution of precipitation. However, these studies did not establish an explicit quantitative relationship between on-line aerosols and IN. Su and

Fung (2018a) implemented the simplified Goddard Chemistry Aerosol Radiation and Transport aerosol model (GOCART) together with Shao's dust emission scheme (Kang et al., 2011; Shao et al., 2011) in WRF/Chem and incorporated the online IN nucleation scheme of DeMott et al. (2015) for producing real time of IN into the double-moment Thompson–Eidhammer microphysics scheme. They analyzed the impact of dust in East

Asia on radiative forcing together with temperature very carefully, and thus only the sensitive impacts in terms of precipitation rate in March and April in 2012 (Su and Fung, 2018b). The spring of 2012 is not a typical dust season, most dust storm concentrated in Mongolia. And their work needs more comparison with real precipitation observations.

In this study, we also employ the Global/Regional Assimilation and PrEdiction System, China Meteorological Administration (CMA) Unified Atmospheric Chemistry Environment (GRAPES/CUACE) model to investigate the effects of dust aerosols on precipitation. GRAPES/CUACE provides on-line sectional aerosol concentrations with multi chemical composition information (Wang et al., 2010; Zhou et al., 2012). Zhou et

al. (2016) introduced an on-line aerosol–CCN–cloud interaction scheme into the system, allowing the model to simulate real time CCN activation and their influence on precipitation. However, in the GRAPES/CUACE microphysics scheme WDM6, IN is a function of temperature, and cloud ice is represented by a single-moment scheme only for the mass mixing ratio (Hong et al., 2004; Hong et al., 2006; Zhang et al., 2022). To

address these limitations, this study implements a double-moment cloud ice scheme and incorporates an on-line aerosol–IN nucleation scheme to explicitly represent heterogeneous processes. Using this improved framework, we then investigate the impact of dust on precipitation by a typical dust affected precipitation event in East



Asia. This paper is organized as follows: Section 2 introduces the model configuration,
cloud microphysical processes, on-line aerosol–IN nucleation scheme, study region,
and observational datasets. Section 3 presents the evaluation of the improved model's
simulation performance and discusses the effects of dust on precipitation. Section 4
summarizes the main conclusions of the study.

## 2 Model description and methodology

### 2.1 GRAPES/CUACE

The GRAPES is a fully compressible, non-hydrostatic numerical weather model
that adopts a semi-implicit and semi-Lagrangian discretization scheme (Chen et al.,
2008; Xu et al., 2008; Zhang and Shen, 2008; Wang et al., 2022a). The physical
packages include cumulus convective, single-moment cloud microphysics, radiative,
land surface, and boundary layer processes. CUACE is an regional chemical weather
forecasting system developed by Gong and Zhang (2008) coupled on-line with
GRAPES (Wang et al., 2010). It is capable of simulating on-line seven aerosol species
of sulfate, nitrate, ammonia, black carbon, organic carbon, sea-salt together with dust
(Zhou et al., 2008, 2012; Wang et al., 2015). The sectional dust emission scheme is by
Marticorena and Bergametti (1995) and Alfaro and Gomes (2001) which has been
improved by surface dust flux observations and desertification in East Asia (Gong et al.,
2003), and new desertification map and soil texture samples from Chinese deserts
(Zhou et al., 2019; Zhou et al., 2024). The aerosol size spectra have been divided into
12 size bins with a radius range of 0.005–0.01, 0.01–0.02, 0.02–0.04, 0.04–0.08, 0.08–
0.16, 0.16–0.32, 0.32–0.64, 0.64–1.28, 1.28–2.56, 2.56–5.12, 5.12–10.24, and 10.24–
20.48 μm. The model has a horizontal resolution of 0.15° and 31 vertical levels
extending to approximately 28.6 km in altitude.

### 2.2 WDM6 microphysics scheme

In this study, we select the WDM6 microphysics scheme in GRAPES for
simulating precipitation (Hong et al., 2004; Hong et al., 2006; Zhang et al., 2022) . The
WDM6 scheme simulates the mass mixing ratio of water vapor (Qv), as well as the
mass and number concentrations of cloud water (Qc) and cloud rain (Qr) in warm



clouds. For icy clouds, it includes the mass mixing ratios of cloud ice (Qi), snow (Qs),
and graupel (Qg). A double-moment cloud ice scheme by Park and Lim (2023) is

incorporated into the WDM6 scheme, allowing for the explicit prediction of cloud ice
number concentration in GRAPES. A sectional CCN activated scheme has been
introduced in GRAPES, connecting the multi-component multi-section aerosols from
CUACE into the WDM6 microphysics and the sub-grid convective parameterization
scheme by newly activated CCN at each time step (Zhou et al., 2016). Thus a fully

aerosol-CCN-cloud interaction scheme has been implemented in GRAPES/CUACE.

### 2.3 On-line aerosol-IN nucleation scheme

In the original WDM6 scheme, when the temperature is below 0 ℃, the production
rate of cloud ice is attributed to two processes: nucleation of ice from vapor ($P_{igen}$) and
deposition-sublimation ($P_{idep}$). The IN concentration is calculated by a classical ice

nuclei nucleation scheme, which is an empirical function of temperature and does not
account for the influence of atmospheric aerosols (Hong et al., 2004; Hong et al., 2006):

$$N_{ice}(m^{-3})=10^3 e^{0.1(T_0-T_k)} \tag{1}$$

Where,$T_k$ is atmospheric temperature,$T_0$ is the freezing point (273.15 K)。

This study aims to implement an on-line aerosol-IN nucleation scheme in CUACE
that accounts for heterogeneous ice nucleation processes influenced by atmospheric

aerosols. Heterogeneous nucleation mechanisms are generally classified into
immersion freezing, condensation freezing, deposition nucleation, and contact freezing
(Hiranuma et al., 2015; Ilotoviz et al., 2016; Lee et al., 2017), the first three of them are
selected which are relatively well developed. And the reasons to choose the three are
also that dust aerosols affect ice nucleation mainly at temperatures below 258.15 K

through the three (Cantrell et al., 2013; Patnaude et al., 2025), and the efficiency of
contact freezing by dust particles is relatively low (Niehaus et al., 2014).

Immersion freezing is a heterogeneous ice nucleation process with existence of
liquid drops at temperatures between 233.15 K and 273.15 K, which ice nucleus
immersed in supercooled liquid, triggering it freezing into an ice crystal (Boose et al.,





2016).The initial size of the ice crystal is influenced by the size of the liquid droplet

(Fan et al., 2014; Gibbons et al., 2018), therefore the cloud ice formation through this

mechanism is relatively easier compared to other nucleation modes. In this study, the

immersion freezing nucleation scheme used is developed by DeMott et al. (2015), based

on continuous flow diffusion chamber measurements. The number concentration of ice

nuclei, $N_{icenui}$ , activated via immersion freezing is given by:

$$N_{icenui}(m^{-3}) = 3 * n_{aer,0.5}^{1.25} * e^{(0.46*(273.16-T_k)-11.6)} \qquad (2)$$

Where, $n_{aer,0.5}^{1.25}$ is the number concentration of insoluble aerosol particles with

diameters exceeding 0.5 μm such as dust, black carbon and part of organic carbon. $\Delta t$

is the integration time step.

Deposition and condensation freezing are both heterogeneous ice nucleation

processes that occur at temperatures between 248.15 K and 258.15 K. In condensation

freezing, water vapor first condenses on the surface of IN and subsequently freezes to

form an ice crystal, while in deposition nucleation, water vapor directly deposits onto

the IN surface (Kanji et al., 2017). The initial size of the ice crystals is comparable to

that of the smallest droplets, and the ice formation through these pathways is generally

harder than that of immersion freezing. In this study, the parameterization scheme

developed by Jiang et al. (2016) is adopted, which was derived from dust events

observed in Xinjiang, Huangshan, and Nanjing in China, using the static vacuum vapor

diffusion chamber Frankfurt Ice nucleation Deposition freezing Experiment. The

number concentration of ice nuclei, $N_{icenud}$ , combines both deposition and

condensation freezing processes into the following:

$$N_{icenud} = 5.7 * 10^{-7} n_{aer,0.5}^{0.018(273.16-Tk)-0.007S_i+0.342} * (273.16 - Tk)^{3.745} * S_i^{1.31} \qquad (3)$$

Where, $S_i$ is supersaturation with respect to ice.

WDM6 uses the formula $\rho q_{I0}(kg\ m^{-3}) = 4.92 \times 10^{-11} N_{ice}^{1.33}$ to calculate

nucleation of ice from vapor due to the IN increase. It ignores the influence of IN size

and heterogeneous ice nucleation processes. In this paper, the relationship between the

IN concentration and the mass concentration of newly generated ice crystals ($q_{Inew}$) is



as follows:

$$\rho q_{Inew}(kg\ m^{-3}) = \frac{4}{3}\pi\rho_i\left(r_{if}^3 \mathrm{N_{icenui}} + r_{df}^3 \mathrm{N_{icenud}}\right) \tag{4}$$

Where, $\rho_i$ is 500 $kg\ m^{-3}$ (Park and Lim, 2023). $r_{if}$ represents the radius of cloud ice formed via immersion freezing, while $r_{df}$ represent the radius of cloud ice formed through deposition and condensation freezing, respectively. The typical range of ice crystal radius in East Asia is about 10–100 μm (Chen et al., 2021), droplet radius range is about 1~30 μm (Um et al., 2018; Yang et al., 2021). Considering ice crystals generally grow from smaller particles and the radius of initial ice crystal size are often smaller than observed values, and with reference to the bin sizes of aerosol particles in CUACE, this study assumes the ice crystal radius of $r_{df}$ and $r_{if}$ to be:

$$\begin{cases} r_{df=10\ \mu m}(r_{aer}<10\ \mu m) \\ r_{df=30\ \mu m}(r_{aer}>10\ \mu m) \end{cases} \tag{5}$$
$$\begin{cases} r_{if=30\ \mu m}\ \ (r_{aer}<10\ \mu m) \\ r_{if=50\ \mu m}\ \ (r_{aer}>10\ \mu m) \end{cases}$$

The mass production rate of cloud ice newly nucleated is calculated using Equation (6):

$$P_{inud}(kgkg^{-1}s^{-1}) = \frac{4}{3}\pi\rho_i\left(r_{df}^3 N_{icenud}\right)/\Delta t \tag{6}$$

$$P_{inui}(kgkg^{-1}s^{-1}) = \frac{4}{3}\pi\rho_i\left(r_{if}^3 N_{icenui}\right)/\Delta t$$

Where, $P_{inud}$ is mass production rate for deposition/condensation freezing,$P_{inui}$ is for immersion freezing.

Then, the original production rate for nucleation of ice from vapor $P_{igen}$ is replaced by the deposition/condensation freezing $P_{inud}$ and immersion freezing $P_{inui}$ described above.

## 2.4 Case description and test design

### Typical dust affected precipitation event

The typical dust affected precipitation event is from 00:00 UTC on 9 April to 00:00 UTC on 15 April 2018, which contains two dust storms events in East Asia. One is from 9 to 11 April, originating in Mongolia affected northern China. Lots of dust storm phenomena are observed in Mongolia, while blowing dust and floating dust phenomena





are reported in central and western Inner Mongolia, central Gansu, Ningxia, northern Shaanxi, most parts of Shanxi, southern Hebei, northern Henan, and western Shandong in China. Another event is from 13 to 14 April. It also gains with widespread dust storm

phenomena in Mongolia and central Inner Mongolia, blowing or floating dust phenomena observed in central Inner Mongolia, northern Shanxi, Beijing, Tianjin, and northern Hebei in China. Between the two dust storm events, the precipitation occurred from west to eat covering most of northern China extending to the Yangtze River area, from 00:00 UTC on 12 April to 00:00 UTC on 15 April, with the highest accumulations

concentrated in Shaanxi, Henan, southern Hebei, and along the Yangtze River in Sichuan, Hubei, Anhui, and the Jiangsu-Zhejiang-Shanghai area.

Figure 1a presents the dust-affected areas by dust phenomenon from Meteorological stations and $PM_{10}$ from the National Environmental Monitoring Network of the Ministry of Environmental Protection. Based on the distribution of dust

in this event, the domain bounded by 90-135 °E and 20-54 °N is defined as the major dust-affected area (DA, outer red rectangle in Figure 1). Together with the real precipitation distribution (Fig. 5a), the domain bounds by 103°–130.5°E and 27.5°– 50°N is defined as the dust-affected precipitation (DP) area (DPA, the inner red rectangle in Figure 1). The whole model domain covers 70°–145°E and 15°–64.5°N,

containing the DA and DPA.

GRAPES/CUACE successfully reproduces both the spatial distribution and intensity of the dust events (Fig. 1b). Considering that many radar observations and model studies have indicated that dust mainly participates in cloud ice processes between 3 and 5 km in altitude (Haarig et al., 2019; He et al., 2021; He et al., 2023),

Fig. 1c also shows the simulated dust within this the 3–5 km altitude range.

**Test design**

As shown in Table 1, three tests are designed. The first test uses the original WDM6 microphysics scheme without considering aerosol effects, denoted as T_CTL. The second test incorporates the on-line aerosol–CCN–cloud interaction scheme from

Zhou et al. (2016), denoted as T_CCN. Based on T_CCN, the third test adds the on-line



aerosol-IN nucleation scheme described in Section 2.3, denoted as T_CCNIN.

The successive integration is cut into several three-days-interval with a warm restart. It starts at 00:00 UTC on April 5, 2018 with 6 days spinning up for tracers in CUACE. Except for water vapor, all initial values of hydrometeors are zero. The outputs are in 3-hour interval. As simulation time increases, integration errors tend to accumulate (Zhang et al., 2019), and to minimize the influence of initial conditions on precipitation, an additional test is conducted from 11 to 13 April. Then the results on April 13 are taken from this test.

The initial and boundary meteorological conditions for GRAPES/CUACE are from the Final Operational Global Analysis data produced jointly by the National Centers for Environmental Prediction (NCEP) and the National Center for Atmospheric Research (NCAR) in a temporal resolution of 6 hours and a spatial resolution of 0.15°. The anthropogenic emissions are from Multi-resolution Emission Inventory for China (Li et al., 2017).

**2.5 Data and evaluation methodology**

Dust observations are obtained from two sources: weather phenomena from the CMA surface meteorological observation network with a temporal resolution of 3 hours, while $PM_{10}$ and $PM_{2.5}$ concentration data from the national environmental monitoring network of the Ministry of Ecology and Environment of China, with a temporal resolution of 1 hour. 6-hour accumulated rainfall data are also from CMA surface meteorological observation network. As there are more than 2,000 precipitation stations in DA, only 63 stations with quality levels 1 and 2 evenly distributed are selected for evaluation, in which 43 stations are in DPA to avoid overfitting with the model outputs. Due to the complex sources of $PM_{10}$ and considering the relative long atmospheric residence time of dust, we select precipitation stations where the $PM_{2.5}/PM_{10}$ ratio is less than 0.6 within 24 hours prior to the precipitation event as representative of dust-influenced precipitation (DP) stations (Wang and Yan, 2007; Filonchyk et al., 2019).

Model performance is evaluated using mean absolute error (MAE), root mean square error (RMSE), and symmetric mean absolute percentage error (sMAPE)



(Shcherbakov et al., 2013):

$$\text{MAE} = \frac{\sum_{i=1}^{n}(r_{mi}-r_{oi})^2}{n} \qquad (8)$$

$$\text{RMSE} = \sqrt{\frac{\sum_{i=1}^{n}(r_{mi}-r_{oi})^2}{n}}$$

$$sMAPE = \frac{1}{n}\sum_{i=1}^{n}\frac{|r_{mi}-r_{oi}|}{|r_{mi}|+|r_{oi}|}$$

$$aMAPE = \frac{r_{mi}-r_{oi}}{|r_{mi}|+|r_{oi}|}$$

where $r_{mi}$ represents the simulated cumulative precipitation at station i, and $r_{oi}$ denotes the observed precipitation. For MAE, RMSE and sMAPE, values closer to 0 indicate better simulation performance. $aMAPE$ $aMAPE = \frac{r_{mi}-r_{oi}}{|r_{mi}|+|r_{oi}|}$ is used to evaluate overestimation and underestimation of the impact. When $aMAPE < 0$,

precipitation is underestimated, and vice versa.

### 3 Results

#### 3.1 Ice nuclei

Figures 2a and 2b show the horizontal distribution of the maximum IN number concentration between 3 and 5 km above ground level in DPA area during 00:00 UTC

on 11 April to 00:00 UTC on 15 April 2018 by T_CTL and T_CCNIN, respectively. Figure 2c presents the vertical distribution of number concentrations of dust with diameters larger than 0.5 μm and IN number concentrations averaged over all DPA stations. Figure 2d shows the vertical distribution of production rate for nucleation of ice. Both the IN number concentration and the production rate for nucleation of ice are

calculated at one model time step (100 s).

The on-line aerosol-IN nucleation scheme can correct the systematic underestimation of IN concentrations. The IN distribution in T_CCN is similar to that in T_CTL, with IN concentrations ranging around $10^0 - 10^1$ L$^{-1}$ between the altitude of 3 and 5 km during the DP event (Fig. 3a), showing a relatively uniform horizontal

pattern. The IN concentration increases with height (Fig. 3c), primarily due to the temperature-dependent nature of original WDM6 scheme. As a result, cloud ice is mainly produced near the −40 ℃ level. Above this layer, IN concentration continues





to increase, but production rate for nucleation of ice begin to decline due to limited cloud water (Fig. 3d). In T_CCNIN, the DP event averaged IN concentrations can reach

$10^3 – 10^4$ L$^{-1}$ and higher near the source area, about 90% from dust aerosl, closer to those observed or simulated in other East Asian dust events (Tobo et al., 2020; Hu et al., 2023; Herbert et al., 2025). The vertical distribution of IN is clearly influenced by both the dust concentration and water/ice saturation (Fig. 3b). At altitudes up to 6 km, both the IN concentration and cloud water decrease (Fig. 3c), and the production rate

for nucleation of ice peaks between 4 and 5 km (Fig. 3d) which is consistent with radar observations and other modeling studies (Haarig et al., 2019; He et al., 2021; He et al., 2023).

### 3.2 Hydrometeors

The on-line aerosol–IN nucleation scheme can modify the distribution of

hydrometeors. Figure 3 shows the event averaged vertical distribution of hydrometeors at DPA stations simulated by T_CTL and T_CCNIN. Figure 4 shows the differences of hydrometeors between T_CTL and T_CCNIN and between T_CCN and T_CCNIN. The hydrometeor variations are further compared across three distinct phases: the 6-hour pre-precipitation (phase 1), the active precipitation period (phase 2), and the 6-

hour post-precipitation (phase 3).

During all phase, dust aerosols suppress the formation of ice-phase hydrometeors within the 0 to −40 ℃ temperature layer. The distribution of ice-phase hydrometeors in T_CCN is similar to that in T_CTL, with ice-phase hydrometeor concentrations ranging around 0.27 - 0.50 g kg⁻¹ within the 0 to −40 ℃ temperature layer during

phase 1~3 (Fig. 3a-f). The highest concentration of ice-phase hydrometeors is in phase 1 (Fig. 3a, d). In T_CCNIN, the mixing ratio of ice-phase hydrometeors decreases to 76-93% of those in T_CCN and T_CTL (Fig. 4a-f). This reduction occurs because below 6 km, the average temperature of T_CCNIN is higher than that of both the T_CCN and T_CTL by about 0.1 to 0.5 °C. This is consistent with other works which

also show that as cloud-top temperatures are higher in dusty conditions, more small-sized ice-phase cloud particles are formed, which could limit ice-phase hydrometeor



development (Huang et al., 2006; Li and Min, 2010).

Within the 0 to -40°C temperature layer, in T_CCNIN, cloud ice mixing ratio decreases by about 0.05 - 0.10 g kg⁻¹ and cloud snow increases by about 0.02 - 0.10 g kg⁻¹, comparing to that of T_CTL and T_CCN during all phase (Fig. 4a-f). This is because as the cloud ice increases, the production rate for accretion of cloud ice by snow is enhanced. As a result, cloud ice is rapidly transformed into snow, leading to higher snow mass concentrations. Below the altitude of 10 km, the mean mass ratio of cloud ice to snow changes from 1: 1 to 1: 3, aligning more closely with observation, which shows that cloud ice generally has higher number concentrations but lower mass concentrations than cloud snow (Gao et al., 2020; Yang et al., 2021; Feng et al., 2021; Fang et al., 2022). This simulated ratio also agrees well with other numerical modeling results (Zhang et al., 2021; Park and Lim, 2023).

During all the three phases, dust aerosols leads to the accumulation of cloud water near the 0 °C level (Fig4 a-f). In the T_CCNIN, cloud water increases by approximately 0.02-0.04 g/kg compared to T_CTL, and by 0.01-0.03 g/kg compared to T_CCN. The increasing dust-IN promotes the formation of cloud ice, which subsequently transforms into cloud water near the 0 °C level, resulting in enhanced cloud water accumulation in DPA, which is consistent with radar-based findings reported by Zhu et al. (2023).

During phase 1, cloud water and rain water in T_CCNIN are reduced by about 0.01 g/kg⁻¹ and during phase 2 and 3, cloud water and rain water in T_CCNIN are reduced by about 0.02 g/kg⁻¹, compared to T_CTL and T_CCN (Fig. 3b,c, e, f). This is because dust-INs compete with CCNs for available water vapor. As production rate for heterogeneous nucleation of cloud ice increases, the development of the precipitation system is suppressed (Wang et al., 2022b; Zhu et al., 2023).

### 3.3 Precipitation

The on-line aerosol–IN nucleation scheme can modulate the spatial distribution of precipitation. Figure 5a shows the observed accumulated precipitation of DPA stations, and Figure 5b shows the simulated accumulated precipitation of T_CTL. In T_CTL, 18 of 43 stations in DPA exhibit overestimated simulation precipitation compared to



observations (overestimated stations), primarily located in areas near dust sources area such as Gansu, Ningxia, Shaanxi, and Inner Mongolia, as well as northeastern provinces including Shandong, Liaoning, Jilin, and Heilongjiang (Fig.5b). At these overestimated stations, the observed mean accumulated precipitation is 9.98 mm, while the simulated

mean accumulated precipitation is 25.55 mm (Fig.6), with an average sMAPE of 42.98 %. The other 25 stations show underestimated simulated precipitation compared to observations (underestimated stations), mainly distributed across Hebei, Beijing, Henan, and the Yangtze River Basin (Fig.5b). At underestimated stations, the observed mean accumulated precipitation is 31.86 mm (Fig.6), while the simulated value is only

5.52 mm, with an average sMAPE of −64.39 %.

In T_CCN, on-line aerosol–CCN–cloud interaction scheme can improve the underestimation of precipitation simulation in areas such as Beijing, Shanxi, Hebei, and Hubei (Fig. 5c). For underestimated stations, mean accumulated precipitation increases by 0.52 mm compared to that of T_CTL (Fig. 6). However, underestimation of

precipitation becomes more severe in Anhui, Jiangsu, Shandong, Sichuan, Chongqing, and parts of Hubei, resulting in no significant improvement in MAE, RMSE and sMAPE (Fig. 7b). For underestimated stations, precipitation simulation improves by approximately 1.57% in T_CCN. For overestimated stations, the simulation performance deteriorated compared to T_CTL; the mean accumulated precipitation is

1.11 mm higher than that in T_CTL (Fig. 6), and precipitation simulation deteriorates by approximately 11%, with MAE increasing by 1.1 and RMSE by 2.1 (Fig. 7a). Overall, precipitation simulation is improved in 22 of 43 stations. It shows that only the influence of CCN by aerosols can introduce some more bias to make those stations performance worse.

In T_CCNIN, the on-line aerosol–IN nucleation scheme does not alter the overall pattern of overestimation precipitation north of 35° N and underestimation precipitation to the south of 35° N in T_CTL (Fig. 5d). However, compared to T_CTL, notable improvements are observed primarily between 34° and 40° N. As discussed in Section 3.2, dust-IN competes with CCN for water vapor at layer with temperature above 0°C,





suppressing precipitation and reducing the overestimation of precipitation near the dust
       source areas. sMAPE is reduced by about 1–10 % in areas near the dust source, resulting
       in more accurate forecasts compared to both T_CTL and T_CCN (Fig. 5e, f).

       Rather than being removed by precipitation or evaporation, the suppressed cloud
       water is transported downstream in T_CCNIN, improving underestimation
precipitation simulations over areas such as Beijing and Shanxi, where sMAPE is
       reduced by 5–77 % (Fig. 5e). Compared with T_CCN, T_CCNIN not only improves
       precipitation simulations between 34° and 40° N, but also shows improvements over
       Sichuan and Hubei. However, it suppresses precipitation over the Yangtze River Basin,
       resulting in increased model simulation error there (Fig. 5f). For underestimation
stations, the mean accumulated precipitation increases by 0.18 mm compared to
       T_CCN, and precipitation simulations improves by approximately 0.6%, with little
       changes in MAE and RMSE (Fig. 6b). For overestimated stations, the mean
       accumulated precipitation decreases by 1.5 mm compared to T_CCN, and precipitation
       simulations improves by approximately 15%, with MAE reduced by 0.8 and RMSE
reduced by 3.2 (Fig. 6a). In all, precipitation simulations at 24 of 43 stations in
       T_CCNIN show improvement compared to both T_CTL and T_CCN.

       In summary, while the on-line aerosol–IN nucleation scheme has limited impact
       on the total precipitation amount, it modulates the spatial and temporal distribution of
       precipitation, same to that of Park and Lim (2023) and Su and Fung (2018b). Dust
aerosols suppress precipitation near source areas; the suppressed cloud water can be
       conserved within the weather system and transported to downwind areas where it can
       enhance the precipitation efficiency there. This redistribution of precipitation improves
       the performance of the GRAPES/CUACE.

## 4 Conclusions and discussion

In order to explore the impact of spring dust aerosols on precipitation, in
       GRAPES/CUACE, this study has improved the WDM6 scheme to predict both cloud
       ice number and mass concentrations. And we also develop an on-line aerosol-IN
       nucleation scheme. The model performance has been evaluated by a typical dust-





precipitation event from 00:00 UTC on 9 April to 00:00 UTC on 15 April 2018.

The on-line aerosol–IN nucleation scheme significantly modifies IN concentration distributions. The original WDM6 scheme exhibits a systematic underestimation of ice nuclei concentrations, with IN concentrations ranging around $10^0 - 10^1 \ L^{-1}$ between 3 and 5 km altitude during the dust-precipitation event, and abnormally increasing vertically due to the temperature-dependent nature of original WDM6

scheme, peaking near the $-40°$ C layer. With the on-line aerosol–IN nucleation scheme, IN concentrations reached $10^3 - 10^4 \ L^{-1}$ between 3 and 5 km altitude, peaking at about the layer between 4 and 5 km in height, and cloud ice is concentrated between 2 and 6 km, which agrees better with observation.

        The on-line aerosol–IN nucleation scheme alters the cloud hydrometeor

distributions. Within the -40 °C to 0°C temperature layer, dust suppresses the formation of ice-phase hydrometeors. The mixing ratio of ice-phase hydrometeors decreases to 76-93% of those in T_CCN and T_CTL. This reduction occurs because during dusty conditions cloud-top temperatures are higher and more small-sized ice-phase cloud particles formed, both of which could limit ice-phase hydrometeor development. The

on-line aerosol–IN nucleation scheme ultimately leads the average ratio of cloud ice to cloud snow changing from 1:1 to 1:3, which is much closer to the observation. This is because that the production rate for accretion of cloud ice by snow also rises as the production rate for nucleation of ice increases, enhancing the accretion of cloud ice to snow.

In the temperature layer above 0°C, increased cloud ice converts to cloud water near the 0 °C temperature layer, resulting in the accumulation of cloud water at layers near the 0 °C temperature layer. Compared to T_CCN, cloud water increases by about 0.02-0.04 g/kg by competing for available water vapor between CCN and IN. The increasing production rate for nucleation of ice suppress the precipitation, leading cloud

water and rain water are reduced by approximately $0.02 \, \mathrm{g\,kg^{-1}}$ during the active precipitation period.

        The on-line aerosol–IN nucleation scheme can also modulate the spatial



distribution of precipitation. In this case, the on-line aerosol–IN nucleation scheme and the on-line aerosol–CCN–cloud interaction scheme does not alter the overall pattern of overestimation precipitation north of 35° N and underestimation precipitation to the south of 35° N as seen in T_CTL. Only considering the influence of CCN by aerosols further increases the bias at DPA stations where precipitation simulation was originally inaccurate.

After comprehensively considering aerosol effects on both IN and CCN, the on-line aerosol–IN nucleation scheme mitigates the overestimation of rainfall in these areas by suppressing precipitation near dust source areas. For overestimated stations, the mean accumulated precipitation decreases by about 1.5 mm compared to T_CCN, with the MAE reduces by 0.8 and the RMSE reduces by 3.2. However, the cloud water suppressed by dust IN is not removed from the atmosphere; instead, it remains in the weather system and releases downstream as the air mass moves, thereby improving the underestimation of precipitation in downstream areas. In stations where precipitation is previously underestimated, the mean accumulated precipitation increases by about 0.18 mm relative to T_CCN.

This study shows comprehensive positive impacts of aerosol on cloud and precipitation by a comprehensive online aerosol-CCN-IN-cloud interaction scheme. It also shows that considering aerosols' impact only on warm cloud-water process can actually increase model inaccuracies, and comprehensively accounting for aerosol effects on both CCN and IN improves precipitation simulations by approximately up to 15 %. As the interactions are influenced by both the aerosols and precipitation weather conditions, more cases in different season and different dusty cases are needed to perform in the near future.

## Code/data availability

All source code and data can be accessed by contacting the corresponding author Chunhong Zhou (zhouch@cma.gov.cn).

## Author contributions.

JZ developed the on-line aerosol-IN nucleation scheme, conducted the data analysis,



and wrote the original draft of this paper. CHZ developed the aerosol-CCN-cloud interaction scheme and the on-line aerosol-IN nucleation scheme, and reviewed and edited the manuscript, providing critical insights. XYS reviewed the manuscript. HW reviewed the manuscript and provided general insight. XYZ reviewed the manuscript and gave guidance on the data analysis. All authors have given approval to the final version of the paper.

**Competing interests**

The authors declare that they have no conflict of interest.

**Financial support.**

This study was jointly supported by the NSFC Project (42090030) and the National Key Project of the Ministry of Science and Technology of China (2022YFC3701205).

**Acknowledgment.**

All figures in this study were produced by the open-source software of MeteoInfoLab from http://www.meteothink.org/index.html. The meteorological initial and boundary conditions for the modeling system were obtained from the China Meteorological Data Sharing Service System (http://data.cma.cn/data/cdcindex/cid/98c64da7ee348b37 html). The meteorological observations were obtained from the China Meteorological Data Sharing Service System (http://data.cma.cn/data/cdcindex/cid/f0fb4b55508804ca. html ). The $PM_{10}$ and $PM_{2.5}$ concentration data from the national environmental monitoring network of the Ministry of Ecology and Environment of China (http://www.cnemc.cn).

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



**Figure**

Figure 1

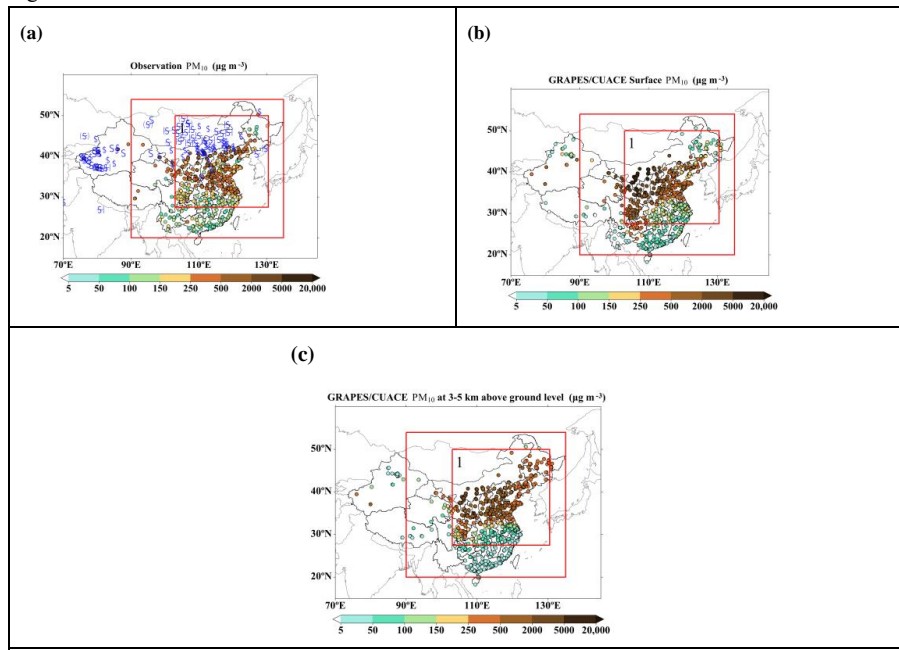

**Figure 1. Observed and simulated affected areas of the dust events from 11 to 15 April. (a) Observed distribution of most sever dusty weather phenomena and maximum PM_{10} concentration for each station during the test time;(b) Same as (a) but for maximum PM_{10} concentration by GRAPES/CUACE;(c) Maximum PM_{10} concentration distribution in the layer of 3-5 km in height by GRAPES/CUACE**





Figure 2

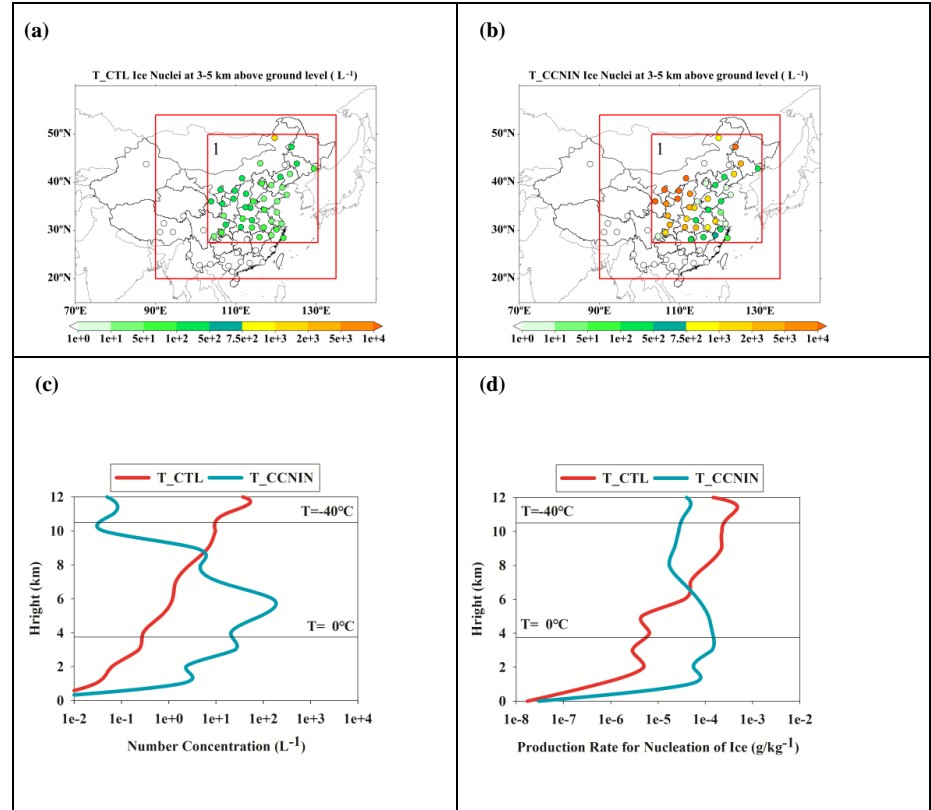

**Figure 2. distribution of IN from 11 to 15 April. (a) Maximum IN number concentration at 3-5 km altitude in T_CTL and T_CCN at DP stations; (b) Same as (a) but for T_CCNIN simulations; (c) Vertical profile of mean IN number concentration in DPA for T_CCN (red line) and T_CCNIN (blue line);(d) Vertical profile of mean production rate for nucleation of ice for T_CCN (red line) and T_CCNIN (blue line)**



Figure 3

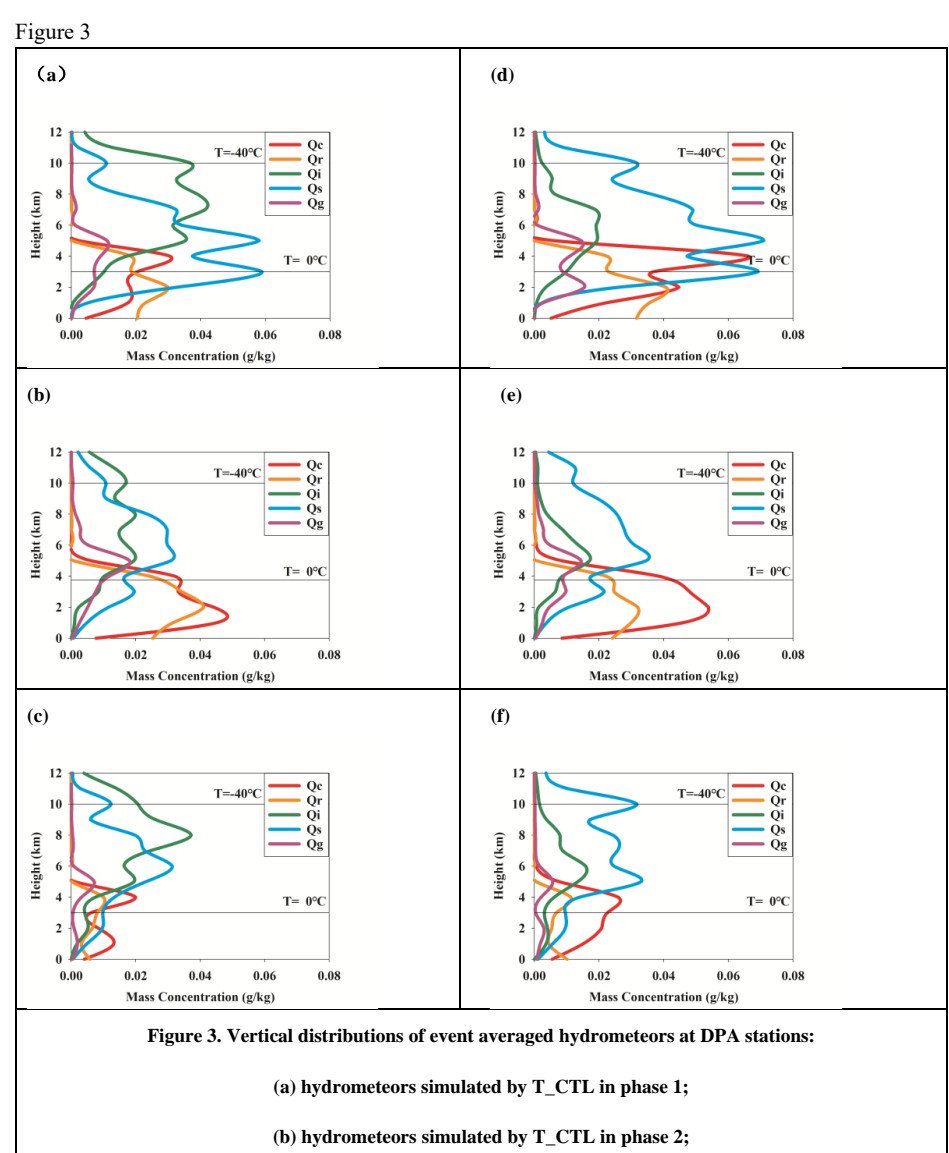

**Figure 3. Vertical distributions of event averaged hydrometeors at DPA stations:**

**(a) hydrometeors simulated by T_CTL in phase 1;**

**(b) hydrometeors simulated by T_CTL in phase 2;**

**(c) hydrometeors simulated by T_CTL in phase 3;**

**(d-f) Same as (a-c) but for T_CCNIN.**




Figure 4

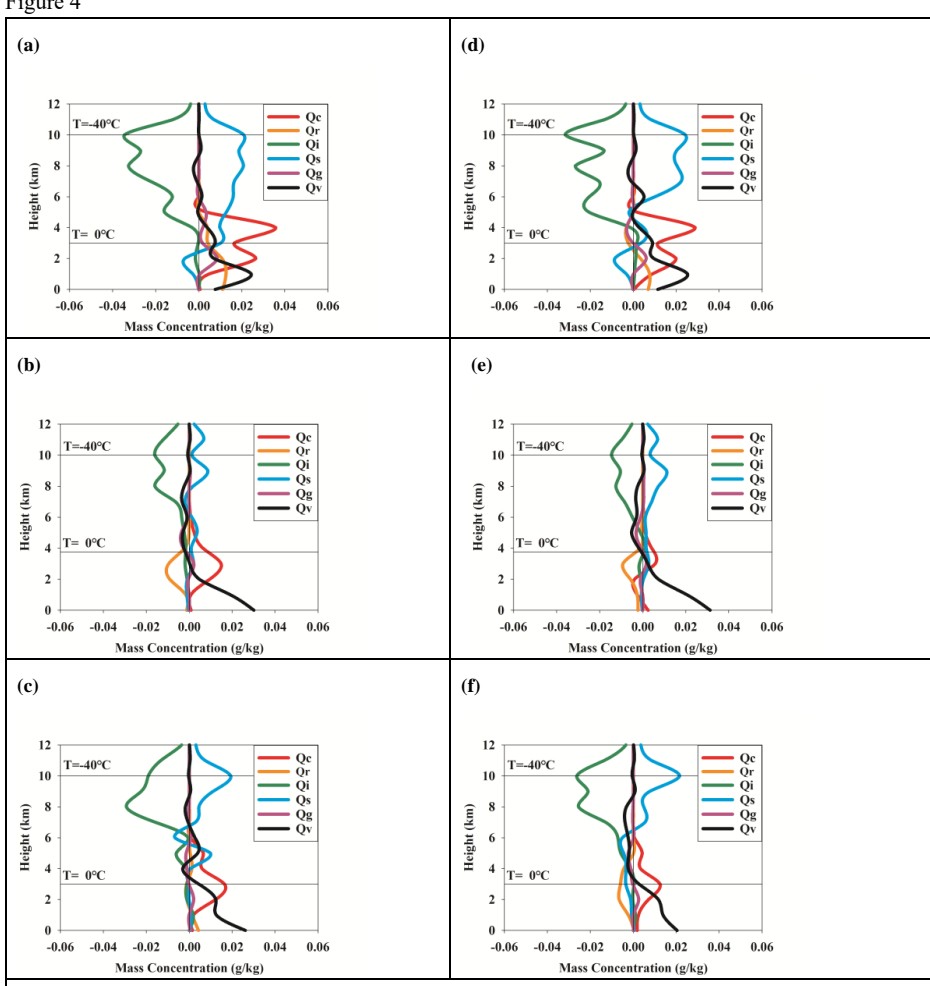

**Figure 4. Differences in event averaged hydrometeor vertical distributions for the dust-precipitation stations:**

**(a) T_CCNIN minus T_CTL in phase 1;**

**(b) T_CCNIN minus T_CTL in phase 2;**

**(c) T_CCNIN minus T_CTL in phase 3;**

**(d-f) Same as (a-c) but for T_CCNIN minus T_CCN.**




Figure 5

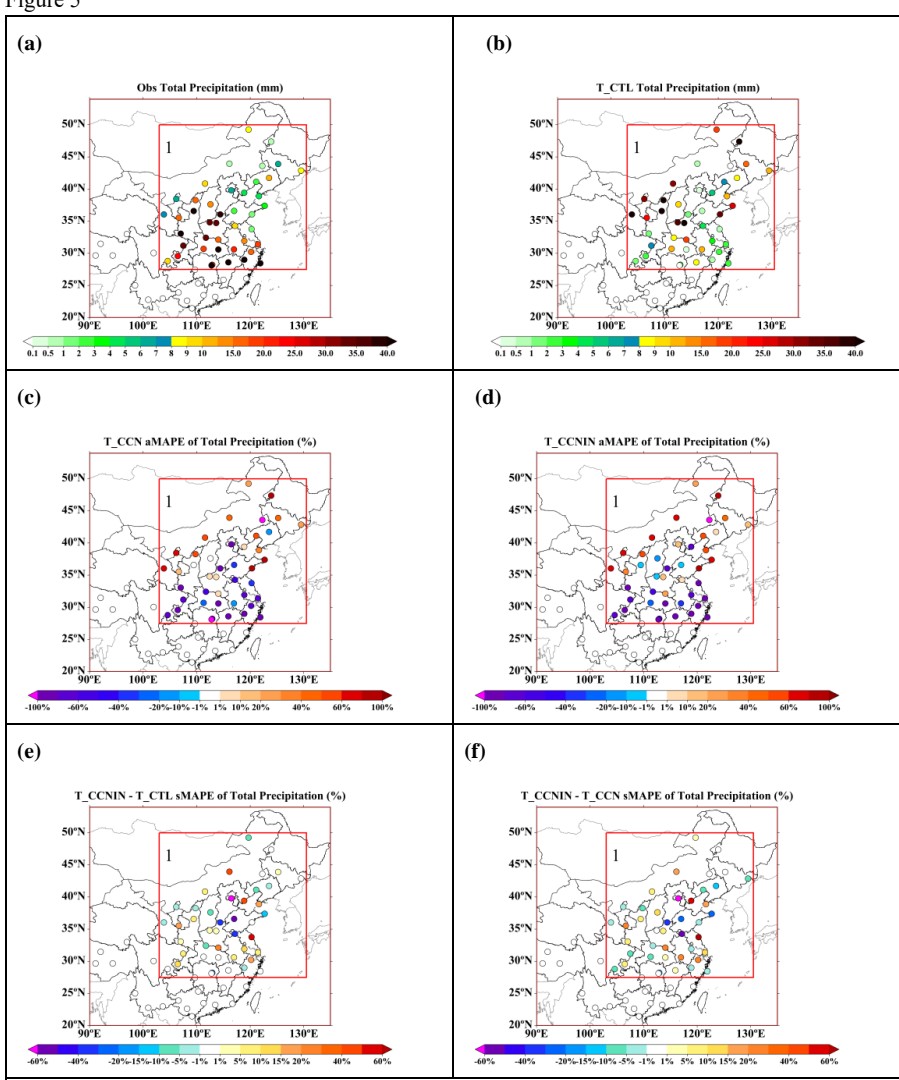

**Figure 5. Comparison of observed and simulated accumulated precipitation at dust-precipitation stations:**

**(a) Observed accumulated precipitation from 11th 00:00 UTC to 15th 00:00 UTC;**

**(b) Same as a but for T_CTL;**

**(c) aMAPE of simulated accumulated precipitation in T_CCN;**

**(d) aMAPE of simulated accumulated precipitation in T_CCNIN;**

**(e) Difference in sMAPE between T_CCNIN and T_CTL;**

**(f) Difference in sMAPE between T_CCNIN and T_CCN.**



Figure 6

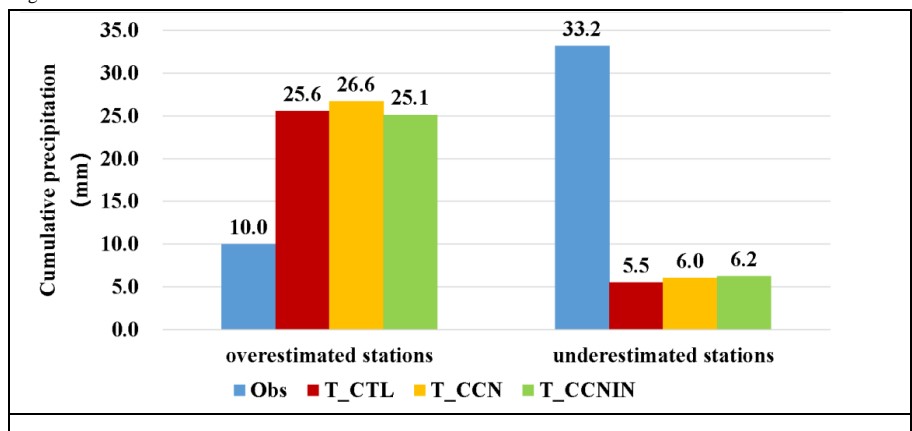

**Figure 6. mean accumulated precipitation during DP event at overestimated stations and underestimated**

**stations**



Figure 7

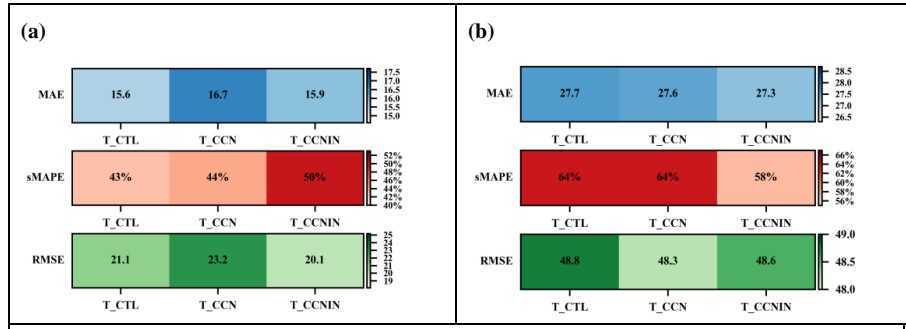

**Figure. 7. Statistical analysis of observed versus simulated accumulated precipitation at DPA stations:**

**(a) Overestimated stations; (b) Underestimated stations.**

**Table**

**Table 1.Three Tests designed for different types of precipitation**

| Test | Warm cloud | Cold cloud |
| --- | --- | --- |
| T_CTL | original WDM6 | original WDM6 |
| T_CCN | on-line aerosol–CCN interaction scheme | original WDM6 |
| T_CCNIN | on-line aerosol–CCN interaction scheme | on-line aerosol-IN nucleation scheme |