# Peer review of "The Impact of aerosol-ice nuclei-cloud interactions on a Typical Spring Dust-Precipitation Event in China"

_EGUsphere, 2025_

## Referee Comment (RC2)

Review for the paper "The Impact of aerosol-ice nuclei-cloud interactions on a Typical Spring Dust-Precipitation Event in China" by Jian Zhang et al.

This study investigates how dust aerosols influence precipitation in China using an improved online aerosol–ice-nucleation (aerosol-IN) scheme implemented in the GRAPES/CUACE regional model. The topic is interesting and important in the field of aerosol–cloud–precipitation interactions. However, in many parts of the manuscript, the authors draw conclusions without sufficient observational evidence. This is the major drawback of the study. Therefore, I am on the negative side regarding publication of this paper.

**General Comments**

**1. Lack of observational analysis:**

As mentioned in the overall assessment, this paper lacks observational analysis to support its conclusions. For example, in lines 317–321, the authors should evaluate model results against radar observations and include water and/or ice saturation information. In line 349, observational evidence for the mass ratio between cloud ice and snow (1:3) should be presented to justify its alignment with observations, as this ratio can vary from case to case. For lines 431–432, there is no analysis or evidence explaining the underestimation of ice nuclei concentrations in the original WDM6 scheme.

**2. Incorrect or missing references:**

In several places, the manuscript either lacks proper references for the physical parameterizations used or cites incorrect previous studies. For instance, *Park and Lim* (2023) and *Kwon et al.* (2023) did not evaluate their results using MODIS observations, contrary to what is stated in line 94. The statement in lines 99–104 is also incorrect. The authors should carefully review and cite previous studies throughout the manuscript. Furthermore, *Hong et al.* (2006) is not the correct reference for the WDM6 scheme (line 155); the appropriate citation is *Lim and Hong* (2010).

**3. Need for microphysical budget analysis:**

To draw reliable conclusions about the vertical profiles of hydrometeors, the authors need to perform a detailed microphysics budget analysis. For example, they argue that increased cloud ice enhances accretion by snow, converting cloud ice to snow. However, this cannot be concluded without budget diagnostics, as other processes—such as aggregation of cloud ice or accretion of snow by rain—could also contribute. I strongly recommend that the authors conduct budget analyses for different stages of precipitation

development.

**4. Aerosol–IN nucleation scheme:**

- The authors should explicitly compare the ice nucleation parameterizations used in *Park and Lim (2023)* and in this study to clarify the differences among T\_CTL, T\_CCN, and T\_CCNIN experiments. A comparative table would be useful. Notably, WDM6 with prognostic cloud ice number concentration (Park and Lim) did not use the formula ρqI0(kg m-3)= 4.92 × 10-11Nice 1.33 to calculate nucleation of ice, which is Pigen, indicating all nucleation processes, in Hong et al., (2024). Instead, Park and Lim's version explicitly treats immersion, contact, and deposition nucleation separately.
- The comparison of IN concentrations between this study and *Park and Lim (2023)* also needs careful reconsideration. Even though the authors replaced the ice nucleation scheme in WDM6 with the prognostic version, they compare IN concentrations with those from older WDM6 versions (*Hong et al., 2024*; *Lim and Hong, 2010*). According to *Eqs. (4–6)* in *Park and Lim (2023)*, IN concentrations are treated differently for contact versus deposition/condensation processes, even though both are temperature-dependent.
- More clarification is needed for their new on-line aerosol-IN nucleation scheme. Does naer,0.5 in Eq.3 represent the same quantity as in Eq. (2)—the number concentration of insoluble aerosol particles larger than 0.5 μm (e.g., dust, black carbon, and some organic carbon)? What is ρ in Eq. (4)?
- 5. Physics parameterization references:

The authors should cite appropriate references for the physics parameterizations used in their GRAPES/CUACE configuration (Section 2.2.1).

6. Support for the downstream transport conclusion:

To substantiate the claim that suppressed cloud water is transported downstream in T\_CCNIN, supportive figures should be presented.

**Specific Comments**

- Line 265:

"As simulation time increases, integration errors tend to accumulate (Zhang et al., 2019), and to minimize the influence of initial conditions on precipitation, an additional test is conducted from 11 to 13 April."

→ This sentence is unclear. Did you perform another simulation starting from 11 April? Please clarify.

**- Line 208:**

"It ignores the influence of IN size and heterogeneous ice nucleation processes." → This is incorrect. The formula includes all heterogeneous ice nucleation processes because Nice represents IN concentration.

- Equations (4) and (6):

These appear to be mathematically identical. Please verify.

- Line 233:

Mark the locations of Ningxia, Shanxi, Hebei, etc., in Figure 1.

- Line 254:

What is the ambient temperature between 3 km and 5 km altitude?

- Figure 2 (page 12):

This should be Figure 3.

- Lines 304–305:

Why do the authors analyze only one model time step (100 s)? What real time does 100 s correspond to?

- Line 306–307:

The authors should show the underestimation of IN concentrations when the online aerosol–IN scheme is not used.

- Line 315:

Besides dust, what other aerosol types serve as IN?

- Lines 329–330:

Specify the actual dates and times corresponding to phases 1, 2, and 3.

- Line 338:

Replace "below 6 km" with "above 4 km."

- Line 340:

Why is the cloud-top temperature higher under dusty conditions?

- Line 350:

Include analysis of hydrometeor number concentrations as well.

- Lines 361–362:

Check this sentence for clarity and correctness.

- Line 365:

Provide further discussion on the precipitation types in the cited papers (*Wang et al.*, 2022; *Zhu et al.*, 2023).

- Lines 426–427:

The statement "this study... mass concentrations" duplicates findings already reported by *Park and Lim* (2023).

---

## Author Comment (AC1)

This study investigates how dust aerosols influence precipitation in China using an improved online aerosol–ice-nucleation (aerosol-IN) scheme implemented in the GRAPES/CUACE regional model. The topic is interesting and important in the field of aerosol–cloud–precipitation interactions. However, in many parts of the manuscript, the authors draw conclusions without sufficient observational evidence. This is the major drawback of the study. Therefore, I am on the negative side regarding publication of this paper.

**General Comments**

Dear editor and reviewers,

Thank you for your thorough review of the manuscript. We have read the reviewer's comments carefully, and have responded and taken your comments into consideration and revised the manuscript accordingly. All the changes have been highlighted in the revised manuscript. Our detailed responses, including a point-by-point response to the reviews and a list of all relevant changes, are as follows:

**1. Lack of observational analysis:**

As mentioned in the overall assessment, this paper lacks observational analysis to support its conclusions. For example, in lines 317–321, the authors should evaluate model results against radar observations and include water and/or ice saturation information. In line 349, observational evidence for the mass ratio between cloud ice and snow (1:3) should be presented to justify its alignment with observations, as this ratio can vary from case to case. For lines 431–432, there is no analysis or evidence explaining the underestimation of ice nuclei concentrations in the original WDM6 scheme.

**A:** Thank you for your careful review of the manuscript. You are correct that direct observations of in-cloud microphysical processes during dust events are extremely difficult to obtain, and many existing studies rely on laboratory experiments to infer microphysical changes under controlled temperature and humidity conditions. In our revision, we have made efforts to validate the model

behavior using available observational evidence from previous field and laboratory studies.

**For example, in lines 317–321, the authors should evaluate model results against radar observations and include water and/or ice saturation information.**

A: Available aircraft and modeling studies indicate that ice-nucleating particles (INPs) and ice-phase hydrometeors tend to concentrate at mid-to-upper tropospheric levels, but their vertical distribution strongly depends on season and thermodynamic conditions, rather than occurring universally near the homogeneous freezing level ($-40\ °C$).

For example, aircraft observations by (He et al., 2023) during non-dust autumn conditions showed peak INP concentrations around 4–5 km, while (Zhang et al., 2021) simulated summertime dust–precipitation in Taiwan interactions and found enhanced ice-phase hydrometeors mainly between 0 and $-20\ ℃$ (approximately 7–8 km).

These studies suggest that the vertical location of maximum ice-phase activity varies substantially across cases. In contrast, the original WDM6 scheme produces a systematic maximum near $-40\ ℃$, which appears inconsistent with both observational and modeling evidence under springtime dust conditions. Our revised scheme shifts the dominant ice-phase production to warmer levels, which is more physically plausible for spring dust–precipitation events.

**in line 349, observational evidence for the mass ratio between cloud ice and snow (1:3) should be presented to justify its alignment with observations, as this ratio can vary from case to case.**

A: Direct observational separation of cloud ice and snow mass is indeed challenging, because radar reflectivity from larger snow particles often masks the

signal from small ice crystals (Kedzuf et al., 2021). Nevertheless, existing aircraft and in situ observations consistently indicate that cloud ice typically has higher number concentrations but lower mass compared with snow (Lawson et al., 2001; Wang et al., 2023). This qualitative relationship is also reproduced in previous modeling studies. This fundamental relationship is reproduced in models, and our chosen ratio of 1:3 is consistent with the range of values reported in prior observationally constrained modeling studies. For instance, Park and Lim (2023)reported a ratio near 1:3, while Zhang et al. (2021) found a ratio close to 1:5 in mid-latitude mixed-phase clouds influenced by dust.

Our simulated ratio (1:3) therefore lies within the range documented by prior observationally constrained studies.

**For lines 431–432, there is no analysis or evidence explaining the underestimation of ice nuclei concentrations in the original WDM6 scheme.**

A: We agree with the reviewer that observational evidence is important for demonstrating the underestimation of IN concentrations in the original WDM6 scheme. Although direct in-cloud INP measurements during dust events are limited, multiple independent field and laboratory studies in East Asia show that dust outbreaks substantially enhance INP concentrations for heterogeneous freezing.

Bi et al. (2019) reported IN concentrations up to 2800 $L^{-1}$during dust-influenced days in May–June 2018 at −20°C to −30 °C using a continuous-flow diffusion chamber in Beijing.

Chen et al. (2021) measured immersion-mode INPs at Peking University Atmosphere Environment Monitoring Station during spring 2018–2019 and found that dust periods increased INP concentrations by approximately two orders of magnitude, reaching $10^{2}L^{-1}$between −15°C and −28°C.

Hu et al. (2023) reported INP concentrations near $10^3 L^{-1}$ at −20 °C at two contrasting northern China sites (a polluted urban site and a clean mountain site), indicating that dust significantly elevates INPs across very different environments.

Measurements of Tobo et al. (2019) at the Tokyo Skytree showed that during transported dust events, immersion-mode IN reached $10^2 L^{-1}$, confirming that dust strongly enhances IN even far downstream of the source.

Across these studies, heterogeneous INP concentrations during East Asian dust events typically fall in the range of $10^2 - 10^3 L^{-1}$ at temperatures.

In contrast, the original WDM6 parameterization produces IN concentrations of only $10^0 - 10^1 L^{-1}$, and shows little distinction between dust and non-dust periods. This mismatch demonstrates that the original WDM6 scheme substantially underestimates heterogeneous IN activation, which is consistent with the reviewer's concern.

**2. Incorrect or missing references:**

**In several places, the manuscript either lacks proper references for the physical parameterizations used or cites incorrect previous studies. For instance, Park and Lim (2023) and Kwon et al. (2023) did not evaluate their results using MODIS observations, contrary to what is stated in line 94. The statement in lines 99–104 is also incorrect. The authors should carefully review and cite previous studies throughout the manuscript. Furthermore, Hong et al. (2006) is not the correct reference for the WDM6 scheme (line 155); the appropriate citation is Lim and Hong (2010).**

A: We have thoroughly re-examined all citations within the manuscript and have corrected the issues pertaining to incorrect or missing references.

In line 99-102:

Moreover, many clouds ice microphysical schemes were single-moment, which only simulated the mass mixing ratio of cloud ice. Such single-moment schemes often led to large biases in cloud ice mass concentrations (Molthan and Colle, 2012; Igel et

al., 2015). In contrast, double-moment ice schemes, which simulate both cloud ice mass and number concentrations, outperform the single-moment schemes in terms of the simulated structure, life cycle, cloud coverage, precipitation, and microphysical properties (Pu and Lin, 2015; Zhao et al., 2021).

In line 110-114:

Park and Lim (2023) develops the revised Weather Research and Forecasting Double-Moment 6-class (WDM6) scheme through the implementation of prognostic cloud ice number concentrations. The excess generation of cloud ice mixing ratio is considerably alleviated.

**3. Need for microphysical budget analysis:**

**To draw reliable conclusions about the vertical profiles of hydrometeors, the authors need to perform a detailed microphysics budget analysis. For example, they argue that increased cloud ice enhances accretion by snow, converting cloud ice to snow. However, this cannot be concluded without budget diagnostics, as other processes—such as aggregation of cloud ice or accretion of snow by rain—could also contribute. I strongly recommend that the authors conduct budget analyses for different stages of precipitation development.**

A: We appreciate the reviewer's suggestion and fully agree that microphysical budget analysis is essential for drawing robust conclusions.

Following this recommendation, we conducted detailed budget diagnostics for cloud ice, snow, cloud water, and rainwater at different vertical layers corresponding to different thermodynamic regimes.

Based on the variation characteristics, the vertical layer is divided into three parts: layer A, above 7 km (temperature below −18 °C); layer B, between 4 and 7 km (temperature approximately −18 °C to -1.5 ℃); and layer C, below 4 km (temperature approximately -1.5 ℃ to 18 ℃).

These budget analyses have been added to the revised manuscript in Section 3.2:

During the DP event, the introduction of the on-line aerosol-IN nucleation scheme allows dust aerosols to alter the distribution of cloud hydrometeors. Figure 3 shows the DP-event-averaged vertical distributions of hydrometeors in T_CTL and T_IN, as well as their difference (T_IN − T_CTL), by using budget analysis. Figure 4 shows the differences in the production rates of different hydrometeors (T_IN − T_CTL).

Cloud ice

In layer A, when dust aerosols are considered, the IN number concentration decreases in T_IN (Fig. 2c), resulting in cloud ice number concentrations in T_IN that are approximately 5 $L^{-1}$ lower than those in T_CTL, about 40% of T_CTL (Fig. 3d). The cloud ice mass concentration is reduced to only 10% – 50% of T_CTL (Fig. 3a,3b). Because the two primary processes contributing to cloud ice formation in this layer—heterogeneous nucleation and deposition-sublimation of cloud ice —are both suppressed (Fig. 4a), and the total production rate of cloud ice (Pigen+Pidep-Psaut-Praci-Psaci-Pgaci) drops to less than 24% of that in T_CTL. On the one hand, the nucleated IN number concentration decreases, weakening the Pigen in T_IN by 1–2 orders of magnitude relative to T_CTL. On the other hand, the reduction in cloud ice number concentration allows the ice crystals to grow more efficiently, with their effective particle size generally reaching 98%-135% of that in T_CTL. The combined effect of these two factors ultimately limits the deposition of water vapor onto the ice crystals. Consequently, Pidep decreases to 20%–50% of T_CTL, with the maximum suppression occurring at approximately 7–8 km (Fig. 4a).

In layer B, cloud ice number concentrations in T_IN range from 7 to 10 $L^{-1}$ , approximately 120% of those in T_CTL. However, the cloud ice mass concentration in T_IN is reduced to only 70%–90% of T_CTL. The effective diameters of cloud ice

also decrease to only 77%–97% of T_CTL, with occasional reductions exceeding 50%. This reduction is mainly attributable to combined effects of enhanced heterogeneous nucleation and suppressed depositional growth, and the total production rate of cloud ice drops to less than 82% of that in T_CTL. Dust aerosols provide additional ice nuclei, leading to a substantial enhancement of heterogeneous nucleation in T_IN and the formation of a much larger number of newly formed small ice crystals, with Pigen exceeding that in T_CTL by more than two orders of magnitude. However, the increase in cloud ice number concentration is accompanied by a reduction in individual particle size, which limits the deposition of water vapor onto ice crystals. As a result, Pidep in T_IN is reduced to about 30% of that in T_CTL, indicating that growth of cloud ice via depositional processes is inhibited.

Snow

In layer A, the total snow production rate in T_IN increases to approximately 88% -200% of that in T_CTL (Psdep+Paacw+Psaut+Piacr+Praci+Psaci+Psacr-Pgaut-Pracs, Fig. 4b), leading to an increase in snow mass concentration to 120%–200% of T_CTL (Fig. 3a, 3b). This increase results from the combined effects of enhanced production rate for deposition-sublimation of snow (Psdep) and weakened production rate for aggregation of cloud ice to snow (Psaut) and production rate for accretion of cloud ice by snow (Psaci). The Psdep can reach approximately 2–5 times that in T_CTL (Fig. 4b). In WDM6, the deposition growth of ice-phase hydrometeors is constrained by the available water vapor, with cloud ice deposition given priority and snow deposition consuming the remaining vapor. Because Pidep is reduced to about 20%–50% of that in T_CTL, more water vapor is allocated to snow deposition, Psdep is then enhanced. Meanwhile, as cloud ice reduces, Psaut and Psaci are weakened in T_IN, with both processes reduced to approximately 40%–60% of their values in T_CTL (Fig. 4a, 4b). Despite the suppression of these source terms, the substantial enhancement of snow deposition growth dominates the snow budget in layer A, resulting in a net increase in snow production and cloud-snow mass concentration.

Finally, the ratio of cloud ice to cloud snow changes from 1:1 to 1:3 in layer A, more closely consistent with observation, which shows that cloud ice generally has higher number concentrations but lower mass concentrations than cloud snow (Gao et al., 2020; Yang et al., 2021; Feng et al., 2021; Fang et al., 2022).

In layer B, the snow mass concentration shows relatively small changes, ranging from approximately 90% to 100% of T_CTL. From the perspective of cloud microphysics, the mechanisms are similar to those in layer A. Despite the reduction of Pidep, the Psdep increases to 130%–200% of T_CTL. At the same time, the decrease in cloud ice mass leads to the continued suppression of Psaut and Psaci, resulting in a total snow production rate of about 95% of T_CTL.

In layer C, although the model diagnostics indicate an enhancement in cloud-snow production processes (production rate for accretion of rain by snow (Psacr) and production rate for accretion of rain by cloud ice (Piacr)) and a reduction in the production rate for accretion of snow by rain (Pracs), newly formed cloud snow cannot be maintained because the temperature is already above 0 °C which makes it instantaneously melt, rapidly converting to rain. As a result, there is no significant change in snow mass concentration in this layer.

Cloud water and rainwater

Cloud water and rainwater are mainly distributed in layer C (temperature approximately -2 °C to 18 °C). In this layer, both cloud-water and rainwater mixing ratios in T_IN are about 90%-95% of those in T_CTL. This small reduction is primarily attributed to a weakening of the production rate for cloud droplet activation from CCN (Pcact), which decreases by about 5% in T_IN relative to T_CTL, indicating a suppressed conversion of water vapor into liquid water. As a consequence of the reduced cloud-water content, the production rate for accretion of rainwater by cloud water (Pracw) is also weakened, by 5%–10%. Meanwhile, the conversion of rainwater into ice-phase hydrometeors (Psaci, Pgaci, and Piaci) is enhanced.

However, under the thermodynamic conditions of layer C, temperatures exceed the melting thresholds of ice-phase hydrometeors, the newly formed snow and graupel rapidly melt and are easily converted back into rainwater. Consequently, these ice-phase conversion processes contribute only marginally to the net change in rainwater mixing ratio.

Overall, dust suppresses cloud development, reducing the total ice-phase hydrometeor content in layer A to 70–85% of T_CTL, the total ice-phase hydrometeor content in layer B to 85–91% of T_CTL, and the liquid-phase hydrometeor content in layer C to 90–95% of T_CTL. Our results indicate that dust aerosols tend to suppress cloud development in springtime dust-related precipitation over East Asia, where precipitation is predominantly stratiform. Similar suppression effects have also been reported in previous observational studies (Wang et al., 2022b; Zhu et al., 2023).

**4. Aerosol–IN nucleation scheme:**

**A:** Thank you for this question. We respond to each point below.

**- The authors should explicitly compare the ice nucleation parameterizations used in Park and Lim (2023) and in this study to clarify the differences among T_CTL, T_CCN, and T_IN experiments. A comparative table would be useful. Notably, WDM6 with prognostic cloud ice number concentration (Park and Lim) did not use the formula $\rho q I_0 (kg\ m-3)$= 4.92 × 10-11$Nice$ 1.33 to calculate nucleation of ice, which is Pigen, indicating all nucleation processes, in Hong et al., (2024). Instead, Park and Lim's version explicitly treats immersion, contact, and deposition nucleation separately.**

A: In GRAPES/CUACE, the original WDM6 scheme follows Hong et al. (2004), in which the ice nucleation of cloud ice is parameterized using

$\rho qI0(kg\ m-3) = 4.92 \times 10\text{-}11 Nice\ 1.33$ which represents the total effect of heterogeneous ice nucleation and is used to diagnose the ice nucleation rate (Pigen). This formulation is still applied in T_CTL.

In contrast, Park and Lim (2023) employed a prognostic ice nucleation framework in which ice nucleation rates are explicitly calculated based on aerosol diffusion coefficients, and different heterogeneous nucleation pathways (immersion, contact, and deposition/condensation freezing) are treated separately. In their formulation, the conversion of production rate of ice nuclei number to production rate of ice mass is expressed as

$$\text{Pinud} = \text{Ninud} \times \frac{4\pi}{3}\frac{\rho_i}{\rho_a}\ Rind^3$$

In this study, the corresponding formulation is

$$P_{inud}(kgkg^{-1}s^{-1}) = \frac{4}{3}\pi\frac{\rho_i}{\rho_a}\ (r_{id}^3 N_{icenud})/\Delta t$$

which is structurally equivalent to that used by Park and Lim (2023). The key difference lies in the treatment of the initial radius of newly nucleated ice crystals. Park and Lim (2023)assumed a fixed initial ice crystal radius of 10 μm, whereas in this study, following Chen et al. (2019), different initial radii are prescribed for immersion freezing and deposition/condensation freezing, reflecting their different growth efficiencies and formation difficulties.

Based on East Asian cloud-ice size observations, we assume:

$$\begin{cases} r_{df=10\ \mu m}(r_{aer}<10\ \mu m) \\ r_{df=30\ \mu m}(r_{aer}>10\ \mu m) \end{cases}$$

$$\begin{cases} r_{if=30\ \mu m}\ (r_{aer}<10\ \mu m) \\ r_{if=50\ \mu m}\ (r_{aer}>10\ \mu m) \end{cases}$$

Therefore, although the formulations differ in implementation details, the core physical linkage between IN number concentration and ice mass production is consistent between this study and Park and Lim (2023).

**- The comparison of IN concentrations between this study and Park and Lim (2023) also needs careful reconsideration. Even though the authors replaced the ice nucleation scheme in WDM6 with the prognostic version, they compare IN concentrations with those from older WDM6 versions (Hong et al., 2024; Lim and Hong, 2010). According to Eqs. (4–6) in Park and Lim (2023), IN concentrations are treated differently for contact versus deposition/condensation processes, even though both are temperature edependent.**

A: In the original WDM6 scheme, the IN concentration is parameterized as

$N_{ice}(m^{-3})=10^3 e^{0.1(T_0-T_k)}$, which represents the total concentration of activated ice nuclei by heterogeneous nucleation. Dividing by the time step ($\Delta t$) yields the heterogeneous nucleation rate (Nigen).

In this study, we diagnose the total heterogeneous nucleation rate as Pigen is diagnosed as the sum of Pinui and Pinud. In T_IN, Nigen is diagnosed as the sum of $N_{icenud}$ and $N_{icenud}$, divided by the time step

Our analysis further shows that immersion freezing is the dominant heterogeneous nucleation mechanism, exceeding deposition and condensation freezing by 4–5 orders of magnitude in DP-event-averaged production rate for nucleated IN number concentration and 5–6 orders of magnitude in production rate of cloud ice. Given this large disparity and for clarity of comparison, we therefore present the heterogeneous nucleation rate (Pigen) consistently in the figures.

**- More clarification is needed for their new on-line aerosol-IN nucleation scheme. Does naer,0.5 in Eq.3 represent the same quantity as in Eq. (2)—the number concentration of insoluble aerosol particles larger than 0.5  μm (e.g., dust, black carbon, and some organic carbon)? What is r in Eq. (4)?**

A: In Eqs. (2) and (3), naer,0.5 represents the number concentration of insoluble aerosol particles larger than 0.5 μm, including dust, black carbon, and a fraction of organic carbon, following Chen et al.

In Eq. (4), r denotes the initial radius of cloud ice formed via different nucleation pathways: $r_{if}$ represents the initial radius of cloud ice formed by immersion freezing, and $r_{df}$ represents the initial radius of cloud ice formed by deposition and condensation freezing.

These values correspond to the assumed initial cloud-ice sizes based on East Asian observations (Um et al., 2018; Chen et al., 2021; Yang et al., 2021).

**5. Physics parameterization references:**

**The authors should cite appropriate references for the physics parameterizations used in their GRAPES/CUACE configuration (Section 2.2.1).**

A: We have thoroughly re-examined all citations within the manuscript and have corrected the issues pertaining to incorrect or missing references.

In line 169-171:

The IN concentration is calculated by a classical ice nuclei nucleation scheme, which is an empirical function of temperature and does not account for the influence of atmospheric aerosols (Hong et al., 2004):

In line 217-223:

The parameterization scheme selected here is developed by Jiang et al. (2016) and (Chen et al., 2019). It first developed by Jiang et al. (2016) based on dust events observed in Xinjiang, Huangshan, and Nanjing in China, using the static vacuum vapor diffusion chamber Frankfurt Ice nucleation Deposition freezing Experiment. Then some parameters of it was refined and extended it to represent both deposition and immersion freezing by Chen et al. (2019) .

**6. Support for the downstream transport conclusion: To substantiate the claim that suppressed cloud water is transported downstream in T_IN, supportive figures should be presented.**

A: Thank you for this question. The statement regarding downstream transport of suppressed cloud water is not inferred directly from Figures 3 or 4, but from an additional diagnostic analysis of hydrometeor fluxes in Section 3.3.

We calculate horizontal hydrometeor fluxes across 116 °E, 33 °–50 °N and 33 °N, 103 °–116 °E from 12:00 UTC on 12 April to 18:00 UTC on 13 April (Fig. 6). Over the entire 0–12 km layer, the total hydrometeor flux slightly increases to about 102% of that in T_CTL.

Within the temperature range from 0 to -40 ℃, the total horizontal hydrometeor flux decreases by about 11 %, primarily due to a substantial reduction in cloud ice flux, accompanied by increases in snow and graupel fluxes. In Layer A, the total hydrometeor flux is about 4.4 $\times$ 10-5 kg s-1, corresponding to about 75 % of T_CTL. Cloud ice flux drops sharply to about 8 % of T_CTL, while snow and graupel fluxes increase markedly to about 19.8 times and 7.8 times, respectively. In Layer B, the total hydrometeor flux is about 2.6 $\times$ 10-6 kg s-1, corresponding to about 93 % of T_CTL, with cloud ice flux reduced to about 28 % of T_CTL, and snow and graupel fluxes increased to about 2.3 times and about 1.8 times, respectively. At temperatures above 0 ℃, the total horizontal hydrometeor flux increases to about 106 % of T_CTL, with cloud water and rainwater fluxes increasing to about 115 % and about 108 %, respectively.

**Specific Comments**

**1.   - Line 265:**

**"As simulation time increases, integration errors tend to accumulate (Zhang et al., 2019), and to minimize the influence of initial conditions on precipitation, an additional test is conducted from 11 to 13 April." → This sentence is unclear. Did you perform another simulation starting from 11 April? Please clarify.**

A: Yes. You are right, we have corrected it.

In line 290-294:

As simulation time increases, integration errors tend to accumulate (Zhang et al., 2019), and to minimize the influence of initial conditions on precipitation, the simulations in this study were divided into several time segments: 5–8 April, 8–11 April, 11–14 April, and 13–16 April. Among these, the simulation results for 13 April were taken from the 11–14 April experiment to minimize the influence of initial conditions on precipitation development.

**2.   - Line 208:**

**"It ignores the influence of IN size and heterogeneous ice nucleation processes."   →**

**This is incorrect. The formula includes all heterogeneous ice nucleation processes because Nice represents IN concentration.**

A: Yes. You are right, Thank you for your comment. The $N_{ice}$ in the original WDM6 scheme represents the total concentration of nucleated ice nuclei by heterogeneous nucleation. Our intended criticism was not that it ignores the existence of heterogeneous nucleation, but rather that it does not differentiate between the various heterogeneous nucleation mechanisms (e.g., immersion, condensation, and deposition freezing) and their potentially distinct impacts on subsequent ice crystal growth.

We have corrected it in line 224 -230:

WDM6 uses the formula $\rho q_{I0}(kg\ m^{-3}) = 4.92 \times 10^{-11} N_{ice}^{1.33}$ and $P_{igen}(kgkg^{-1}s^{-1}) = \frac{(q_{I0}-q_I)}{\Delta t}$ to calculate newly nucleation of ice. Where, ρ denotes the newly-formed air density, and $q_{I0}$ is the predicted ice mixing ratio (kg kg-1). Δt is the integration time step. Production rate for heterogeneous nucleation is calculated as the difference between $q_{I0}$ and the current ice mixing ratio ($q_I$). However, it does not account for the influence of nucleated IN size or the specific characteristics of different heterogeneous ice nucleation mechanisms on ice crystal development.

**3.   - Equations (4) and (6):**

**These appear to be mathematically identical. Please verify.**

A: Yes,We have corrected this in the revised manuscript. The redundant equation (previously labeled as Equation (6)) has been removed entirely.

We have corrected it in line 231-235:

In this paper, The mass production rate of cloud ice newly nucleated is calculated using Equation (4):

$$P_{inud}(kgkg^{-1}s^{-1}) = \frac{4}{3}\pi\frac{\rho_i}{\rho_a}\left(r_{df}^3 N_{icenud}\right)/\Delta t \tag{4}$$

$$P_{inui}(kgkg^{-1}s^{-1}) = \frac{4}{3}\pi\frac{\rho_i}{\rho_a}\left(r_{if}^3 N_{icenui}\right)/\Delta t$$

Where, Where, $P_{inud}$ is mass production rate for deposition/condensation freezing,$P_{inui}$ is for immersion freezing. $P_{inud}$ depletes water vapor to form cloud ice, while $P_{inui}$ depletes cloud water to form cloud ice.

4. **- Line 233:**

**Mark the locations of Ningxia, Shanxi, Hebei, etc., in Figure 1.**

A: Yes. We have added Supplementary Figure 1 to show the locations of Ningxia, Shanxi, Hebei, and other relevant provinces.

[Figure]

Fig. S1    China Province Map, where B denotes Beijing; T denotes Tianjin; H denotes Hong Kong.

**5.   - Line 254:**

**What is the ambient temperature between 3 km and 5 km altitude?**

A: Thank you for this important question. The ambient temperatures provided below are the area- and time-averaged values from our model simulation, specifically averaged over the Dust-Precipitation station and during the Dust-Precipitation period. Based on the vertical levels of our GRAPES/CUACE configuration, the interpolated ambient temperatures at the specified altitudes are as follows:

3 km: 275.33 K

4 km: 271.91 K

5 km: 265.30 K

**6.   - Figure 2 (page 12):**

**This should be Figure 3.**

A: Yes. You are right, we have corrected it.

7.  **Why do the authors analyze only one model time step (100 s)? What real time does 100 s correspond to?**

A: Thank you for this important question. The value of 100 s refers specifically to the internal integration time step of the GRAPES model's dynamical core. We recognized that explicitly stating this numerical parameter was unnecessary for interpreting the results of the ice nucleation rate and cloud ice growth rate. In the revised manuscript, we no longer analyze microphysical tendencies over a single model integration time step. Instead, all microphysical processes (heterogeneous nucleation (Pigen) and production rate for deposition- sublimation rate of cloud ice (Pidep)) are expressed consistently in terms of physical rates (e.g., $g\ kg^{-1}s^{-}$), which are independent of the model's internal integration time step.

8.  **- Line 306–307:**

**The authors should show the underestimation of IN concentrations when the online aerosol–IN scheme is not used.**

A: Thank you for this important question. Figures 2a and 2b show the maximum nucleated    IN number concentration during the dust–precipitation event, while Figure 2c shows the event-averaged vertical distribution. Multiple field and laboratory studies confirm that the nucleated IN concentrations simulated by our improved scheme are consistent with observational and experimental results:

Bi et al. (2019) reported IN concentrations up to $2800\ L^{-1}$ during dust-influenced days in May–June 2018 at $-20°C$ to $-30\ °C$ using a continuous-flow diffusion chamber.

Chen et al. (2021) measured immersion-mode INPs at PKUERS during spring 2018–2019 and found that dust periods increased INP concentrations by approximately two orders of magnitude, reaching $10^2\ L^{-1}$ between $-15°C$ and $-28°C$.

Hu et al. (2023) reported INP concentrations near $10^3$ L$^{-1}$ at $-20$ °C at two contrasting northern China sites (a polluted urban site and a clean mountain site), indicating that dust significantly elevates INPs across very different environments.

Measurements of Tobo et al. (2019) at the Tokyo Skytree showed that during transported dust events, immersion-mode IN reached $10^2$ L$^{-1}$, confirming that dust strongly enhances IN even far downstream of the source.

In contrast, the original WDM6-based scheme (T_CTL) produces nearly uniform and much lower IN concentrations (around $10^0$-$10^1$ L$^{-1}$), indicating a systematic underestimation of dust-related IN activity.

We have corrected it in line 239-244:

The on-line aerosol-IN nucleation scheme can correct the systematic underestimation of IN concentrations. The maximum nucleated IN number concentrations in T_CTL can reach $10^2$ L-1 in layer B during the DP event (Fig. 2a), showing a relatively uniform horizontal pattern, which is much lower than observed IN concentrations ($10^2$–$10^4$ L$^{-1}$) during East Asian dust events (Bi et al., 2019; Tobo et al., 2019; Chen et al., 2021; Hu et al., 2023).

**9. - Line 315: Besides dust, what other aerosol types serve as IN?**

A: Thank you for this important question. In addition to dust aerosols, our model also considers black carbon (BC) and a fraction of organic carbon (OC) aerosols to serve as ice-nucleating particles (INPs), following the approach of Chen et al. (2019).

**10. - Lines 329–330:**

**Specify the actual dates and times corresponding to phases 1, 2, and 3.**

A: Thank you for this important question.

Because the occurrence time of dust–precipitation coupling varies among observation stations across area 1, we adopt a condition-based compositing approach, focusing on periods when dust and precipitation coexist. Specifically, only time steps and grid points that satisfy $PM_{2.5}/PM_{10} < 0.6$ and surface precipitation greater than 0.1 mm are included in the analysis. This approach allows us to robustly quantify the average microphysical responses to dust across different locations and times.

We have revised the manuscript to clarify that the phases refer to event-based, regional-time averages, not fixed simulation periods in Line 311-315.

Due to the complex sources of $PM_{10}$ and considering the relatively long atmospheric residence time of dust, we select precipitation stations where the $PM_{2.5}/PM_{10}$ ratio is less than 0.6 within 24 hours prior to the precipitation event as representative of dust-influenced precipitation (DP) stations (Wang and Yan, 2007; Filonchyk et al., 2019).

11. **- Line 338:**

   **Replace "below 6 km" with "above 4 km."**

   A: Yes. You are right, we have corrected it.

12. **- Line 340:**

   **Why is the cloud-top temperature higher under dusty conditions?**

   A: We analyzed the microphysical budget terms and the associated temperature tendencies in the cloud layer.

The budget analysis shows that, in the 5–8 km layer, several ice-phase microphysical processes are modified under dusty conditions. Specifically, the conversion of water vapor and rainwater to cloud ice is suppressed, with the corresponding growth rates reduced to less than 50% of those in T_CCN. In contrast,

the conversion of water vapor to snow and graupel is enhanced, reaching approximately 2–3 times that in T_CCN.

However, when these microphysical changes are translated into temperature tendencies, their net contribution to temperature change remains very small. The event-averaged temperature tendency induced by microphysical processes is on the order of $-0.002$ to $-0.001$ °C in T_CCN, and about 70% of that magnitude in T_IN. Although these changes may contribute marginally to a slightly higher cloud-top temperature, their magnitude is far too small to be considered the primary driver of the cloud-top temperature increase.

Therefore, we conclude that the higher cloud-top temperature under dusty conditions is not mainly controlled by cloud microphysical processes, but likely reflects the combined effects of large-scale thermodynamic conditions and dynamical adjustments, while microphysical effects play only a secondary role. Finally, the discussion on cloud-top temperature has been removed from this paper.

**13. - Line 350:**

**Include analysis of hydrometeor number concentrations as well.**

A: Yes. In the revised manuscript, we have included an explicit analysis of hydrometeor number concentrations by taking advantage of the double-moment cloud-ice representation. The corresponding results are presented in Section 3.2.

Above 7 km, when dust aerosols are considered, the IN number concentration decreases in T_IN (Fig. 2c), leading to cloud-ice number concentrations that are approximately 5 $L^{-1}$ lower than those in T_CTL, corresponding to about 40% of T_CTL (Fig. 3d). Meanwhile, the cloud-ice mass concentration is reduced more substantially, to only 10%–50% of that in T_CTL (Fig. 3a,b).

In contrast, within the 4–6 km layer, dust aerosols provide additional ice-nucleating particles, increasing cloud-ice number concentrations in T_IN to 7–10 $L^{-1}$, which is about 120% of T_CTL. However, the cloud-ice mass concentration in this layer is still reduced to 70%–90% of T_CTL. Consistently, the effective diameter of cloud ice decreases to 77%–97% of that in T_CTL, with occasional reductions exceeding 50%.

These results indicate that changes in cloud-ice mass are not solely controlled by number concentration, but also strongly modulated by particle growth efficiency.

**14. - Lines 361–362:**

**Check this sentence for clarity and correctness.**

A: Following the reviewer's suggestion, we have substantially revised the analysis by (1) increasing the model output frequency from 3 h to 1 h and (2) introducing a detailed microphysical budget analysis focused on dust–precipitation periods. As a result, the analytical framework of Section 3 has changed from a comparison between pre-dust and post-dust stages to an event-averaged analysis during dust–precipitation occurrences.

Because the original statement in Lines 361–362 was based on the previous analysis framework and is no longer consistent with the revised methodology and results, we have removed this sentence from the manuscript.

**15. - Line 365:**

**Provide further discussion on the precipitation types in the cited papers (Wang et al., 2022; Zhu et al., 2023).**

A: Thank you for this suggestion. We have clarified the precipitation regimes discussed in the cited studies.

Zhu et al., (2023) investigated the impacts of dust aerosols on both convective and stratiform precipitation over southeastern China during summer, and showed that under dusty conditions, precipitation tends to intensify in the mid-troposphere (−5 to +2 ℃) but weakens in both upper and lower layers, indicating an overall suppression of precipitation efficiency.

Wang et al. (2022b) is a review study. Rather than repeating its general conclusions, we directly discuss several observational studies summarized therein. For example, Min et al. (2009) analyzed a stratiform precipitation event associated with a trans-Atlantic Saharan dust outbreak and found that dust aerosols weakened precipitation intensity. Hui et al. (2008) showed that over the West African Sahel, dust aerosols suppress stratiform rainfall by increasing CCN concentrations and modifying the boundary-layer thermodynamic structure.

Overall, although these studies differ in region and season, they mainly involve stratiform precipitation systems. This is consistent with our study, which focuses on springtime dust–frontal precipitation over East Asia, where precipitation is predominantly stratiform. Therefore, these studies provide relevant observational support for our conclusions.

In Line 450-453:

Our results indicate that dust aerosols tend to suppress springtime dust-related precipitation over East Asia, where precipitation is predominantly stratiform. Similar suppression effects have also been reported in previous observational studies (Wang et al., 2022b; Zhu et al., 2023).

**16.  - Lines 426–427:**

**The statement "this study... mass concentrations" duplicates findings already reported by Park and Lim (2023).**

**A:** Yes. We have corrected it.

We have removed the redundant statement and revised the text in Lines 510–513:

In order to explore the impact of spring dust aerosols on precipitation, in GRAPES/CUACE, we develop an on-line aerosol-IN nucleation scheme. The model performance has been evaluated by a typical dust-precipitation event from 00:00 UTC on 9 April to 00:00 UTC on 15 April 2018.

**Related References**

Bi, K., McMeeking, G. R., Ding, D. P., Levin, E. J. T., DeMott, P. J., Zhao, D. L., Wang, F., Liu, Q., Tian, P., Ma, X. C., Chen, Y. B., Huang, M. Y., Zhang, H. L., Gordon, T. D., and Chen, P.: Measurements of Ice Nucleating Particles in Beijing, China, Journal of Geophysical Research: Atmospheres, 124, 8065–8075, https://doi.org/10.1029/2019JD030609, 2019.

Chen, J., Wu, Z., Chen, J., Reicher, N., Fang, X., Rudich, Y., and Hu, M.: Size-resolved atmospheric ice-nucleating particles during East Asian dust events, Atmospheric Chemistry and Physics, 21, 3491–3506, https://doi.org/10.5194/acp-21-3491-2021, 2021.

Chen, Q., Yin, Y., Jiang, H., Chu, Z., Xue, L., Shi, R., Zhang, X., and Chen, J.: The Roles of Mineral Dust as Cloud Condensation Nuclei and Ice Nuclei During the Evolution of a Hail Storm, Journal of Geophysical Research Atmospheres, 124, https://doi.org/10.1029/2019JD031403, 2019.

Fang, W., Lou, X., Zhang, X., and Fu, Y.: Numerical Simulations of Cloud Number Concentration and Ice Nuclei Influence on Cloud Processes and Seeding Effects, Atmosphere, 13, 1792, https://doi.org/10.3390/atmos13111792, 2022.

Feng, Q., Niu, S., Niu, T., Fan, X., Shen, D., and Yang, J.: Aircraft—Based Observation of the Physical Characteristics of Snowfall Cloud in Shanxi Province, Chinese Journal of Atmospheric Sciences (in Chinese), 45, 1146–1160, 2021.

Filonchyk, M., Yan, H., Shareef, T. M. E., and Yang, S.: Aerosol contamination survey during dust storm process in Northwestern China using ground, satellite observations and atmospheric modeling data, Theor Appl Climatol, 135, 119–133, https://doi.org/10.1007/s00704-017-2362-8, 2019.

Gao, Q., Guo, X., He, H., Liu, X., Huang, M., and Ma, X.: Numerical Simulation Study on the Microphysical Characteristics of Stratiform Clouds with Embedded

Convections in Northern China based on Aircraft Measurements, Chinese
Journal of Atmospheric Sciences (in Chinese), 44, 899–912, 2020.

He, C., Yin, Y., Huang, Y., Kuang, X., Cui, Y., Chen, K., Jiang, H., Kiselev, A.,
Möhler, O., and Schrod, J.: The Vertical Distribution of Ice-Nucleating Particles
over the North China Plain: A Case of Cold Front Passage, Remote Sensing, 15,
4989, https://doi.org/10.3390/rs15204989, 2023.

Hong, S.-Y., Dudhia, J., and Chen, S.-H.: A Revised Approach to Ice Microphysical
Processes for the Bulk Parameterization of Clouds and Precipitation, Monthly
Weather Review, 132, 103–120, https://doi.org/10.1175/1520-
0493(2004)132%253C0103:ARATIM%253E2.0.CO;2, 2004.

Hu, Y., Tian, P., Huang, M., Bi, K., Schneider, J., Umo, N. S., Ullmerich, N., Höhler,
K., Jing, X., Xue, H., Ding, D., Liu, Y., Leisner, T., and Möhler, O.:
Characteristics of ice-nucleating particles in Beijing during spring: A comparison
study of measurements between the suburban and a nearby mountain area,
Atmospheric Environment, 293, 119451,
https://doi.org/10.1016/j.atmosenv.2022.119451, 2023.

Hui, W. J., Cook, B. I., Ravi, S., Fuentes, J. D., and D'Odorico, P.: Dust-rainfall
feedbacks in the west african sahel, Water Resour. Res., 44,
https://doi.org/10.1029/2008WR006885, 2008.

Igel, A. L., Igel, M. R., and Heever, S. C. van den: Make It a Double? Sobering
Results from Simulations Using Single-Moment Microphysics Schemes,
https://doi.org/10.1175/JAS-D-14-0107.1, 2015.

Jiang, H., Yin, Y., Wang, X., Gao, R., Yuan, L., Chen, K., and Shan, Y.: The
measurement and parameterization of ice nucleating particles in different
backgrounds of China, Atmospheric Research, 181, 72–80,
https://doi.org/10.1016/j.atmosres.2016.06.013, 2016.

Kedzuf, N. J., Chiu, J. C., Chandrasekar, V., Biswas, S., Joshil, S. S., Lu, Y., van
Leeuwen, P. J., Westbrook, C., Blanchard, Y., and O'Shea, S.: Retrieving
microphysical properties of concurrent pristine ice and snow using polarimetric
radar observations, Atmos. Meas. Tech., 14, 6885–6904,
https://doi.org/10.5194/amt-14-6885-2021, 2021.

Lawson, R. P., Baker, B. A., Schmitt, C. G., and Jensen, T. L.: An overview of
microphysical properties of arctic clouds observed in May and July 1998 during
FIRE ACE, J. Geophys. Res.: Atmos., 106, 14989–15014,
https://doi.org/10.1029/2000JD900789, 2001.

Min, Q.-L., Li, R., Lin, B., Joseph, E., Wang, S., Hu, Y., Morris, V., and Chang, F.: Evidence of mineral dust altering cloud microphysics and precipitation, Atmos. Chem. Phys., 9, 3223–3231, https://doi.org/10.5194/acp-9-3223-2009, 2009.

Molthan, A. L. and Colle, B. A.: Comparisons of Single- and Double-Moment Microphysics Schemes in the Simulation of a Synoptic-Scale Snowfall Event, https://doi.org/10.1175/MWR-D-11-00292.1, 2012.

Park, S.-Y. and Lim, K.-S. S.: Implementation of Prognostic Cloud Ice Number Concentrations for the Weather Research and Forecasting (WRF) Double-Moment 6-Class (WDM6) Microphysics Scheme, Journal of Advances in Modeling Earth Systems, 15, e2022MS003009, https://doi.org/10.1029/2022MS003009, 2023.

Pu, Z. and Lin, C.: Evaluation of double-moment representation of ice hydrometeors in bulk microphysical parameterization: comparison between WRF numerical simulations and UND-Citation data during MC3E, Geosci. Lett., 2, 11, https://doi.org/10.1186/s40562-015-0028-x, 2015.

Tobo, Y., Adachi, K., DeMott, P. J., Hill, T. C. J., Hamilton, D. S., Mahowald, N. M., Nagatsuka, N., Ohata, S., Uetake, J., Kondo, Y., and Koike, M.: Glacially sourced dust as a potentially significant source of ice nucleating particles, Nat. Geosci., 12, 253–258, https://doi.org/10.1038/s41561-019-0314-x, 2019.

Um, J., McFarquhar, G. M., Stith, J. L., Jung, C. H., Lee, S. S., Lee, J. Y., Shin, Y., Lee, Y. G., Yang, Y. I., Yum, S. S., Kim, B.-G., Cha, J. W., and Ko, A.-R.: Microphysical characteristics of frozen droplet aggregates from deep convective clouds, Atmospheric Chemistry and Physics, 18, 16915–16930, https://doi.org/10.5194/acp-18-16915-2018, 2018.

Wang, Y. and Yan, Z.: Effect of Different Verification Schemes on Precipitation Verification and Assessment Conclusion, Meteorological Monthly, 33, 9 ( 53-61 ), https://doi.org/10.3969/j.issn.1000-0526.2007.12.008, 2007.

Wang, Y., Kong, R., Cai, M., Zhou, Y., Song, C., Liu, S., Li, Q., Chen, H., and Zhao, C.: High small ice concentration in stratiform clouds over Eastern China based on aircraft observations: Habit properties and potential roles of secondary ice production, Atmospheric Research, 281, 106495, https://doi.org/10.1016/j.atmosres.2022.106495, 2023.

Wang, Z., Xue, L., Liu, J., Ding, K., Lou, S., Ding, A., Wang, J., and Huang, X.: Roles of Atmospheric Aerosols in Extreme Meteorological Events: a Systematic Review, Curr Pollution Rep, 8, 177–188, https://doi.org/10.1007/s40726-022-00216-9, 2022.

Yang, J., Hu, X., Lei, H., Duan, Y., Lv, F., and Zhao, L.: Airborne Observations of Microphysical Characteristics of Stratiform Cloud Over Eastern Side of Taihang Mountains, Chinese Journal of Atmospheric Sciences, 45(1), 88–106, 2021.

Zhang, M., Yu, H., Guo, J., Shen, X., Su, Y., Xue, H., and Dou, B.: Assessment on Unsystematic Errors of GRAPES_GFS 2.0, Journal of Applied Meteorological Science, 30, 332–344, 2019.

Zhang, Y., Yu, F., Luo, G., Fan, J., and Liu, S.: Impacts of long-range-transported mineral dust on summertime convective cloud and precipitation: a case study over the Taiwan region, Atmospheric Chemistry and Physics, 21, 17433–17451, https://doi.org/10.5194/acp-21-17433-2021, 2021.

Zhao, X., Lin, Y., Luo, Y., Qian, Q., Liu, X., Liu, X., and Colle, B. A.: A Double-Moment SBU-YLIN Cloud Microphysics Scheme and Its Impact on a Squall Line Simulation, Journal of Advances in Modeling Earth Systems, 13, e2021MS002545, https://doi.org/10.1029/2021MS002545, 2021.

Zhu, H., Li, R., Yang, S., Zhao, C., Jiang, Z., and Huang, C.: The impacts of dust aerosol and convective available potential energy on precipitation vertical structure in southeastern China as seen from multisource observations, Atmospheric Chemistry and Physics, 23, 2421–2437, https://doi.org/10.5194/acp-23-2421-2023, 2023.

---

## Author Comment (AC2)

**Dear editor and reviewers,**

Thank you for your thorough review of the manuscript. We have read the reviewer's comments carefully, and have responded and taken your comments into consideration and revised the manuscript accordingly. All the changes have been highlighted in the revised manuscript. Our detailed responses, including a point-by-point response to the reviews and a list of all relevant changes, are as follows:

**1.      The logic contradiction. The discussions related to Figures 3 and 4 mentioned the reduction of mass concentration of ice crystal compared between T_IN and T_CCN (or between T_CCN and T_CTL), while the IN nucleation rate is increased in T_IN run. Why does enhanced IN nucleation lead to reduced mass concentration of ice crystal? However, in some part of the text, the authors stated "more ice-phase cloud particles" (L442-444) and "cloud ice increases" (L345-347). The descriptions about the IN effects on ice microphysics are inconsistent in the manuscript, which makes me confused about the conclusions.**

A: Thank you for your careful review of the manuscript. We agree that the original manuscript did not clearly distinguish between changes in cloud-ice number concentration and mass concentration, which led to confusion in the interpretation of the ice-nucleation effects. We have carefully revised the manuscript and re-examined the cloud-ice budget to clarify this issue.

Our updated analysis shows that the impacts of dust-related ice nucleation on cloud ice differ across temperature layers.

Above 7 km (temperature below −17 °C), the dust leads to a reduction in IN number concentrations. As a result, both production rate for heterogeneous nucleation (Pigen) and production rate for deposition- sublimation rate of cloud ice (Pidep) are suppressed, leading to decreases in both cloud-ice number concentration and mass concentration.

In contrast, between 4 and 7 km (temperature approximately −17 °C to 2 °C), the increase in ice nuclei enhances heterogeneous ice nucleation in T_IN, thereby increasing the cloud-ice number concentration. However, the total cloud-ice mass concentration is reduced. This is because the increase in ice crystal number leads to a decrease in the effective particle size, reducing it to 77%-96% of that in T_CTL which limits deposition- sublimation rate of cloud ice. Consequently, although more ice crystals are formed, their individual growth is suppressed, resulting in a reduction in cloud-ice mass.

We have revised the related text (e.g., Section 3.2 and 4) to explicitly distinguish between cloud-ice number concentration and mass concentration and to remove ambiguous expressions such as "cloud ice increases" without qualification. These revisions ensure a physically consistent interpretation of the IN effects on ice microphysics throughout the manuscript.

2.      The heterogeneous ice nucleation scheme. The authors mentioned at L201-202 that the parameterization of Jiang et al. (2016) was derived from dust events in Xinjiang, Huangshan, and Nanjing, China. But in the Introduction part, the authors cited from the same literature (Jiang et al., 2016) and wrote that the dust events were only observed in Huangshan and Nanjing. Please check the correct location of the dust events in the literature (Jiang et al., 2016). Moreover, if the authors are interested in the effects of dust particles on precipitation, why not chose an ice nucleation parametrization that developed base on the dust events occurred only over Xinjiang, which is much closer to the dust source area? As I know, Huangshan and Nanjing are not the major source of the dust events. The atmospheric condition of Huangshan is relatively clean and the main aerosol source of Nanjing could be the polluted aerosols. Even the dust event may pass through these places, the dust particles may be mixed with the local emitted aerosols (lie polluted ones) and make the ice nucleation being complicated.

A: Thank you for pointing out this issue. We have carefully rechecked Jiang et al. (2016), and the observation sites in their study indeed include Xinjiang, Huangshan, and Nanjing. The description in the Introduction has been corrected accordingly.

In Jiang et al. (2016), aerosol samples were collected during both dust and non-dust periods at all three sites using a newly developed high-voltage electrostatic aerosol collector (HVEAC). Ice nucleation was subsequently simulated with a static vacuum water-vapor diffusion chamber to derive immersion-freezing parameterizations. The study provided three site-specific IN parameterizations as well as a general parameterization applicable across all regions, with the differences mainly lying in the empirical coefficients.

Regarding the choice of the parameterization used in our model, we adopt the scheme of Chen et al. (2019). Chen et al. (2019) refined the coefficients of the original Jiang et al. (2016) formulation and extended it to represent both deposition and immersion freezing.

Therefore, we chose the Chen et al. (2019) version of the parameterization rather than using only the Xinjiang-specific coefficients.

The manuscript has been revised in line 216-221:

The parameterization scheme selected here is developed by Jiang et al. (2016) and (Chen et al., 2019). It first developed by Jiang et al. (2016) based on dust events observed in Xinjiang, Huangshan, and Nanjing in China, using the static vacuum vapor diffusion chamber Frankfurt Ice nucleation Deposition freezing Experiment. Then some parameters of it was refined and extended it to represent both deposition and immersion freezing by Chen et al. (2019) .

**3.** **The horizontal resolution of 0.15º is too coarse for investigation of IN effects on precipitation from aspect of microphysics. And the output interval of 3 h is too long to analyze the IN nucleation and related microphysical processes.**

A: Thank you for this valuable comment. We agree that a horizontal resolution of 0.15 ° is relatively coarse for resolving detailed ice-nucleation–microphysics interactions. However, our study aims to investigate the regional-scale precipitation response to dust over East Asia. Similar or even coarser resolutions have been widely used in previous studies focusing on dust–precipitation interactions, for example, Zhang et al. (2021) using GEOS-Chem at 2 ° × 2.5 ° with 1-hourly output, and Luo et al. (2023) using WRF v4.0 at 1 ° × 1 ° with 6-hourly output.

Regarding the temporal output frequency, we appreciate your suggestion. In the original setup, we used a 3-hour output interval because the observational precipitation dataset from the China Meteorological Administration (CMA) provides 6-hour accumulated precipitation. Following your recommendation, we have rerun the simulations with 1-hour output intervals, and the 1-hour fields are now aggregated to match the 6-hour accumulated precipitation observations used for evaluation. This modification allows us to better resolve the temporal sequence of heterogeneous nucleation, cloud microphysical evolution. The manuscript has been revised accordingly.

The manuscript has been revised in line 295-296:

The model outputs 1-hourly precipitation data. To compare with the observed 6-hourly precipitation, the model outputs are temporally interpolated to the time stamps of the observations.

**4.** **The English gramma of the manuscript should be improved.**

A: Thank you for pointing out the need for language improvement. We have thoroughly revised the entire manuscript to enhance its clarity and grammatical accuracy.

such as line 70-95:

Compared with the relatively well-understood impacts of aerosols as CCN, the role of dust as IN is considerably more complex and remains poorly understood, with substantial uncertainties (Kaufman et al., 2002; Eastwood et al., 2008; Pan et al., 2017; Possner et al., 2017). Observational studies have reported diverse and sometimes contradictory relationships between dust and precipitation, depending on temporal scale, season, and environmental conditions. Temporal scale and seasonality play a critical role in shaping the observed relationships between dust and precipitation. At interannual scales, Han et al. (2008) found a significant negative correlation between dust storm frequency and precipitation over the Taklimakan Desert, whereas a positive correlation was observed at monthly scales, suggesting that dust–precipitation relationships are scale dependent. Seasonal contrasts have also been reported. Using long-term ground-based observations, Wang (2013) showed that dust aerosols tend to suppress precipitation over arid and semi-arid regions in spring but may enhance precipitation in summer. In addition to temporal variability, the impacts of dust on clouds and precipitation also exhibit strong regional and environmental dependence. In contrast, Naeger (2018) found that dust could enhance precipitation over Florida based on multi-sensor satellite observations and field campaigns. More recently, Hu et al. (2023) demonstrated that the impact of springtime dust on precipitation is strongly modulated by the presence of other aerosol types. Liu et al. (2024) analyzed the spatiotemporal patterns and trends of dust aerosols and precipitation and found that dust increases suppress precipitation over source regions such as the Gobi and Taklamakan deserts, but enhance precipitation in downwind areas like northern China. Overall, due to the multiple factors influencing precipitation beyond aerosols, it remains challenging to quantify the impact of dust on precipitation from observations alone (Zhou et al., 2016; Stier et al., 2024), highlighting the need for process-oriented numerical modeling studies with physically based aerosol–ice nucleation parameterizations.

**Specific Comments**

1.      Section 2: The authors mentioned that "The aerosol size spectra have been divided into 12 size bins" (L148-149), and they also noticed that WDM6 scheme is a double moment (bulk) 6-class microphysics. I would like to know how a size-resolved aerosol scheme being coupled with a bulk microphysics?

   A: Thank you for the question. Although WDM6 is a bulk double-moment scheme, the heterogeneous ice nucleation process uses the size-resolved aerosol information from CUACE. Aerosols are predicted in 12 sectional bins. For heterogeneous nucleation, the IN number concentration is first computed for each relevant bin (bins 5–12) using Equation (2) and (3) as the aerosol diameter is large than 0.5 $\mu$m in these bins. The number for each bin are then summed to obtain the grid-scale heterogeneous number production rate. And mass production rate of IN for each bin are then summed to obtain the grid-scale heterogeneous mass production rate. They are passed to the bulk WDM6 microphysics. Thus, the aerosol process is size-resolved, while the microphysical tendencies remain bulk, and the coupling is achieved through aggregation of all aerosol-bin contributions.

2.      Section 2.3: How are the advection of IN (that are not nucleated into ice crystals yet) with winds considered in the T_IN test?

   A: Thank you for the question. The aerosol particles predicted by CUACE are transported by the model's dynamical core through advection, diffusion, and sedimentation. At each time step, the heterogeneous ice nucleation rate is diagnosed locally from the transported aerosol fields together with the ambient temperature and supersaturation. Therefore, unactivated aerosols are then advected downstream by the winds, and whether they can serve as IN in downstream regions depends entirely on the local thermodynamic conditions there.

3.      L23-24: I don't understand how the on-line aerosol-IN nucleation scheme modifies the density of IN.

A: Thank you for pointing this out. In our model, the "density of IN" mentioned here refers to the concentration of activated ice nuclei, which is diagnosed within the microphysics scheme rather than being a transported aerosol variable.

The original WDM6 scheme and our on-line aerosol–IN nucleation scheme both compute the activated IN concentration using Equation (1)–(3). Therefore, changing the nucleation parameterization modifies the fraction of aerosols that are activated as IN, even though the aerosol concentration remains unchanged. This is why the on-line aerosol-IN nucleation scheme modifies the concentration of IN.

The manuscript has been revised in line 24-28:

Dust modifies the spatial distribution and number concentration of IN, affecting heterogeneous ice nucleation. Compared with the systematic underestimation in original WDM6, the peak values of nucleated INs can reach $10^{-4}$ $L^{-1}$ with the improved scheme, which is closer to observations. Cloud ice is reasonably formed at altitudes between 4 and 7 km in height.

4.      L24-25: What is the definition of the number concentration of IN? Is that the number of dust particles with diameter exceeding a threshold (like 0.5 μm)? Or is that the number of dust particles that can be activated into ice crystals under appropriate temperature and saturation conditions based on equations 2 and 3? And from which figure did the authors reach the conclusion that "INs reach $10^3$-$10^4$ $L^{-1}$ with the improved scheme"? I did not find the related discussions about the magnitude of IN number.

A: Thank you for this important question. In this study, the number concentration of ice nuclei (IN) does not refer simply to the number of dust particles exceeding a fixed size threshold (e.g., 0.5 μm).   Instead, the IN number concentration is defined as the number of dust aerosol particles that have been activated into ice crystals under the local temperature and supersaturation conditions, as diagnosed by the heterogeneous ice nucleation parameterization based on equations 2 and 3.

Regarding the magnitude of IN concentrations, the spatial distribution and vertical profiles of nucleated IN number concentration are presented in Figures 2a–c. Figures 2a and 2b show the maximum nucleated IN number concentration during the dust–precipitation event, while Figure 2c shows the event-averaged vertical distribution. With the improved on-line aerosol–IN nucleation scheme (T_IN), the maximum IN concentrations can locally reach $10^3$–$10^4$ $L^{-1}$ between 3 and 5 km altitude during intense dust influence, whereas the event-mean values are generally lower. We have clarified this distinction between maximum and averaged IN concentrations in the Section 3.1 in revised manuscript to avoid confusion.

5. **L29-30: "the ratio of cloud ice to cloud snow": Is that "the ratio of mass concentration of ice crystal to that of snow"?**

A: Yes, the ratio of cloud ice to cloud snow is the ratio of mass concentration of ice crystal to that of snow. We have therefore removed this sentence from the manuscript to avoid confusion.

6. **L30-31: It is difficult to understand this sentence.**

A: Thank you for pointing this out. You are correct that the original statement is unclear and inconsistent with the revised results. After reanalyzing the cloud microphysical processes and updating the model outputs, this description is no longer valid. We have therefore removed this sentence from the manuscript to avoid confusion.

7. **L31-33: Please explain why "rainwater is decreased due to vapor competition between IN and cloud condensation nuclei".**

A: Thank you for pointing this out. We agree that the original explanation stating was overly simplified and potentially misleading. We have therefore re-examined this issue using a detailed budget analysis and revised the explanation accordingly.

Our updated analysis shows that the reduction in rainwater in T_IN is not caused by a direct competition for water vapor between IN and CCN. Instead, it primarily results from changes in cloud microphysical pathways induced by dust aerosols. Specifically, the introduction of the online aerosol–IN nucleation scheme enhances heterogeneous ice nucleation in between 4 and 7 km (temperature approximately −17 ℃ to 2 ℃), leading to an increase in the number of small ice crystals.

And the warm-cloud condensation process (Pcact) below 4 km (temperature approximately -2 ℃ to 18 ℃) is slightly weakened, reducing cloud-water production. The subsequent autoconversion and accretion from cloud water to rainwater (Pracw) are also reduced. Therefore, both cloud-water and rainwater mixing ratios in T_IN are 90%-95% of those in T_CTL by approximately

The manuscript has been revised in line 35-37:

Below 4 km, dust suppresses the conversion of water vapor to cloud water and of cloud water to rain, reducing the liquid-phase hydrometeor content to 90–95% of T_CTL.

**8.     L60-62: Are there dust events in Huangshan and Nanjing, China?**

**A:** Thank you for pointing this out. Jiang et al. (2016) performed ice-nuclei measurements at three sites in China: Huangshan, Nanjing, and Xinjiang. Their observations showed that IN concentrations at all three sites were significantly higher during dust-influenced periods than during non-dust conditions.

In line 64-66:

Jiang et al. (2016) found that IN concentrations observed during dust events in Huangshan, Xinjiang and Nanjing were significantly higher than those during non-dust conditions.

**9.      L69-73: Please explain why the relationship between precipitation events and dust storm frequency shows a significant negative correlation at interannual scales but a positive correlation at monthly scales.**

**A:** Thank you for the comment. The opposite correlations at different time scales have also been reported in previous observational studies. The key reason is that dust influences precipitation through different mechanisms at event scale versus climate (interannual) scale (Han et al., 2008).

At the event or monthly scale, dust storms are often accompanied by weak precipitation events. Although the precipitation amount per event is typically small (often less than 1 mm), the occurrence of dust storms and light precipitation events tends to coincide, which produces a positive correlation at the monthly scale.

However, at the interannual scale, the long-term presence of abundant dust aerosols over the Taklimakan Desert substantially increases the number of cloud condensation nuclei. This causes cloud water to be distributed over more droplets, reducing cloud droplet effective diameters and suppressing warm-rain formation. As a result, years with more frequent dust storms generally experience reduced precipitation, yielding a negative interannual correlation.

Thus, the positive correlation at monthly scales reflects the co-occurrence of dust storms with weak precipitation events, while the negative correlation at interannual scales reflects the long-term suppression of precipitation efficiency by excessive dust loading.

**10.      L74-76: Please explain why "dust aerosols tend to suppress precipitation over arid and semi-arid areas in spring, while promoting it in summer".**

**A:** Thank you for the comment. The opposite effects of dust aerosols on precipitation in spring and summer over arid and semi-arid regions have been reported

in previous observational studies (Wang, 2013). Satellite analyses show that the key difference arises from the seasonal changes in cloud optical properties and cloud microphysics.

In spring, clouds over these regions generally have relatively small optical thickness and low liquid water content. Dust loading is negatively correlated with cloud optical depth and cloud water path, indicating that the absorbing dust heats the cloud layer and reduces cloud water, thereby suppressing cloud development and precipitation. Cloud effective diameters also tends to increase, which further weakens warm-rain processes.

In summer, clouds are optically thicker and contain more condensate. Dust is positively correlated with cloud optical depth and cloud water path, suggesting that dust aerosols may enhance cloud growth under the moister and more convectively nucleated summer conditions, leading to increased precipitation. These contrasting relationships explain why dust tends to suppress precipitation in spring but enhance it in summer.

**11.      L67-84: I did not find a clear logic in this paragraph. Please improve it.**

A: Yes, we have revised the corresponding description in Lines 70–95 as follows:

Compared with the relatively well-understood impacts of aerosols as CCN, the role of dust as IN is considerably more complex and remains poorly understood, with substantial uncertainties (Kaufman et al., 2002; Pan et al., 2017). Observational studies have reported diverse and sometimes contradictory relationships between dust and precipitation, depending on temporal scale, season, and environmental conditions. Temporal scale and seasonality play a critical role in shaping the observed relationships between dust and precipitation. At interannual scales, Han et al. (2008) found a significant negative correlation between dust storm frequency and precipitation over the Taklimakan Desert, whereas a positive correlation was observed at monthly scales, suggesting that dust–precipitation relationships are scale

dependent. Seasonal contrasts have also been reported. Using long-term ground-based observations, Wang (2013) showed that dust aerosols tend to suppress precipitation over arid and semi-arid regions in spring but may enhance precipitation in summer. In addition to temporal variability, the impacts of dust on clouds and precipitation also exhibit strong regional and environmental dependence. Case-based and regional observational studies further highlight the complexity of dust–cloud–precipitation interactions. Satellite and aircraft measurements by Rosenfeld and Bell (2011) indicated that dust aerosols reduce cloud droplet effective diameters and precipitation efficiency without significantly changing total cloud water content. In contrast, Naeger (2018) found that dust could enhance precipitation over Florida based on multi-sensor satellite observations and field campaigns. More recently, Hu et al. (2023) demonstrated that the impact of springtime dust on precipitation is strongly modulated by the presence of other aerosol types. Overall, due to the multiple factors influencing precipitation beyond aerosols, it remains challenging to quantify the impact of dust on precipitation from observations alone (Zhou et al., 2016; Stier et al., 2024), highlighting the need for process-oriented numerical modeling studies with physically based aerosol–ice nucleation parameterizations.

**12.    L112-114: I don't understand what this sentence is used for.**

A: Yes, thank you for pointing this out. We cited Su and Fung (2018b) to illustrate that previous studies have attempted to explore dust–precipitation linkages in spring, but often under relatively weak dust conditions. In contrast, our study focuses on a typical spring dust–precipitation event, and places particular emphasis on the cloud microphysical pathways, especially the role of dust as ice-nucleating particles, together with direct comparisons to precipitation observations. The text has been revised to clarify this distinction.

The relevant descriptions have been added to the manuscript (Lines 122–128) to improve clarity:

The spring of 2012 is not a typical dust season, most dust storm concentrated in Mongolia. Therefore, the microphysical pathways through which dust affects precipitation during typical dust events remain insufficiently studied. In this study, we want to focus on the in influence of typical dust storm on precipitation. In contrast, this study focuses on a representative spring dust–precipitation event and explicitly examines the cloud microphysical processes associated with dust-induced heterogeneous ice nucleation, together with direct comparisons to precipitation observations.

**13.    L151-152: The horizontal resolution is 0.15º for the simulation in the present study. It is too coarse to investigate the IN effects on cloud and precipitation from the aspect of microphysics. Can the authors conduct the additional runs with fine resolution to see if the results are robust?**

**A:** Thank you for the suggestion. We agree that using a finer horizontal resolution would provide more detailed representations of cloud microphysical structures. The resolution of the current version of GRAPES/CUACE is consistent with the meteorology, aerosol and gas chemistry together with the emissions. In the future, we would try our hard to improve the resolution. But currently, we have to use this resolution.

And, the objective of the present study is to investigate the regional-scale effects of dust-induced heterogeneous ice nucleation on precipitation over East Asia rather than the storm-scale cloud microphysics. For this purpose, the $0.15°$ resolution used in GRAPES/CUACE is consistent with many previous regional studies of aerosol–cloud–precipitation interactions, including those examining dust impacts (e.g., (Zhang et al., 2021), at $2° \times 2.5°$). At this scale, the model is able to capture the large-scale transport of dust aerosols, their interaction with clouds, and the resulting precipitation response.

**14.**     **L167-169: I don't understand this sentence. In the original WDM6 scheme, is only the nucleation of ice from vapor considered (like condensation freezing and deposition nucleation)? How about the other ice nucleation schemes, like immersion freezing, contact freezing, and homogeneous freezing?**

A: Thank you for raising this important question. In the original WDM6 scheme, the number concentration of ice nuclei is diagnosed solely as a function of temperature, and the production rate for heterogeneous nucleation (Pigen) is computed accordingly. In the original WDM6 scheme, heterogeneous nucleation consumes water vapor to form cloud ice.

After introducing the on-line aerosol–IN nucleation scheme, heterogeneous ice nucleation processes are explicitly distinguished. Specifically, Pinud represents deposition and condensation freezing, which consume water vapor to form cloud ice; Pinui represents immersion freezing, which consumes cloud water to form cloud ice. These two components together constitute Pigen in the updated On-line aerosol-IN nucleation scheme. Homogeneous freezing is treated separately following the original WDM6 formulation, while contact freezing is not explicitly modified in this study.

The relevant descriptions have been added to the manuscript (Lines 45–46) to improve clarity.

In the original WDM6 scheme, when the temperature is below $0\ ℃$, the production rate of cloud ice is attributed to two processes: heterogeneous nucleation (Pigen) and deposition- sublimation rate of cloud ice (Pidep). Both consume water vapor to form ice clouds.

In Lines 234 -235:

$P_{inud}$ depletes water vapor to form cloud ice, while $P_{inui}$ depletes cloud water to form cloud ice.

**15.    L169-172: Did the nucleation of IN consume water vapor in the original WDM6 scheme? I didn't find the effect of water vapor on the ice nucleation from equation (1).**

A: Thank you for this valuable comment. In the original WDM6 scheme, Equation (1) lack explicit water vapor dependency. The reviewer is correct that Eq. (1) solely calculates the nucleated ice nuclei number concentration based on temperature. The conversion of this ice nuclei number to heterogeneous ice nucleation rate and the associated water vapor consumption are handled in subsequent steps of the scheme, as expressed by:

$$q_{I0}(kg \ m^{-3}) = 4.92 \times 10^{-11} N_{ice}^{1.33}$$

$$P_{igen}(kg/kg/s) = \frac{(q_{I0} - q_I)}{\Delta t}$$

Where, $q_{I0}$ represents the cloud-ice mixing ratio (kg/kg), $q_I$ is the existing cloud-ice mixing ratio (kg/kg), $\rho$ is the air density, and $\Delta t$ is the model time step (100 s). During each time step, the water vapor mixing ratio decreases by $P_{igen} \times \Delta t$, while the cloud-ice mixing ratio increases by the $P_{igen} \times \Delta t$. Therefore, although water vapor does not explicitly appear in Eq. (1), the vapor depletion is explicitly accounted for in the microphysics budget of the WDM6 scheme through the Pigen process.

In line 178-181:

In the original WDM6 scheme, when the temperature is below 0 ℃, the production rate of cloud ice is attributed to two processes: heterogeneous nucleation (Pigen) and deposition-sublimation rate of cloud ice (Pidep). Both consume water vapor to form ice clouds.

**16.    L178-180: This sentence is difficult to understand.**

A:Yes, We have revised the original writing In line 188-196:

Heterogeneous nucleation mechanisms are generally classified into immersion freezing, condensation freezing, deposition nucleation, and contact freezing (Hiranuma et al., 2015; Ilotoviz

et al., 2016; Lee et al., 2017). Among these mechanisms, immersion freezing, condensation freezing, and deposition nucleation are selected, as they are relatively well developed. This selection is based on the fact that dust aerosols primarily affect ice nucleation at temperatures below 258.15 K through these three mechanisms (Cantrell et al., 2013; Patnaude et al., 2025), whereas the efficiency of contact freezing by dust particles is relatively low (Niehaus et al., 2014).

**17.  L192-193: There is no "$\Delta t$" in equation (2).**

A:Thank you for the question. Indeed, in the original formulation of DeMott et al. (2015), Equation (2) does not include a time-step term ($\Delta t$).

Within the original WDM6 microphysics scheme, this conversion from IN number concentration to IN nucleation rate is required to couple the parameterization into the prognostic microphysics equations. The rate of heterogeneous ice nucleation is expressed as:

$$N_{inui}(m^{-3}s^{-1}) = \mathrm{N_{icenui}}/\Delta t$$

where $\Delta t$ is the model time step (100 s). Therefore, Equation (2) in our paper directly follows the functional form of DeMott et al. (2015) for calculating $N_{inui}$.

**18.  L194-195: Why do the deposition nucleation and condensation freezing only occur at temperature between 248.15 K and 258.15 K. This temperature range seems too narrow.**

A: Thank you for the question. The implementation of these nucleation schemes in our model, including the specific temperature range of 248.15 K to 258.15 K for deposition nucleation and condensation freezing, follows the work of (Chen et al., 2019). In their study, Chen et al. adopted and applied the parameterization schemes from (Jiang et al., 2016) for deposition nucleation and condensation freezing and (DeMott et al., 2017) for immersion freezing, specifically to simulate dust-hail interactions in East Asia. The temperature range in question is from the original Jiang et al. (2016) scheme, which was based on their observational analysis.

We have revised the corresponding description in Lines 209–220 as follows:

Deposition and condensation freezing are both heterogeneous ice nucleation processes that occur at temperatures between 248.15 K and 258.15 K (Chen et al., 2019).

The parameterization scheme selected here is developed by Jiang et al. (2016) and (Chen et al., 2019). It first developed by Jiang et al. (2016) based on dust events observed in Xinjiang, Huangshan, and Nanjing in China, using the static vacuum vapor diffusion chamber Frankfurt Ice nucleation Deposition freezing Experiment. Then some parameters of it was refined and extended it to represent both deposition and immersion freezing by Chen et al. (2019) .

19.    L198-200: Please provide the evidence for this sentence.

A: Thank you for question. This study references the findings of Chen et al. (2019). For immersion freezing, the size of an initial ice crystal is influenced by the size of the droplet from which it forms; specifically, the initial ice crystal size corresponds to the droplet size. For Deposition and condensation freezing, the process initiates from the smallest droplet size bin. However, since the GRAPES/CUACE model does not employ a bin microphysics scheme for droplets, this study distinguishes between these two freezing mechanisms by setting a small initial size for the cloud ice particles generated via this pathway.

Moreover, DeMott et al. (2015) have demonstrated that immersion freezing is the predominant mode of heterogeneous nucleation in the atmosphere, whereas deposition and condensation freezing are relatively more difficult to occur.

We have revised the corresponding description in Lines 213–216 as follows:

The initial size of the ice crystals is comparable to that of the smallest droplets (Chen et al., 2019), and the ice formation through these two pathways is generally harder than that through immersion freezing (DeMott et al., 2015).

**20.    L207: What are the physical meaning for "ρ" and "qI0"?**

A: Thank you for the question. ρ denotes the air density (kg m$^{-3}$), and q_I0 is the predicted ice mixing ratio after accounting for newly formed ice from heterogeneous nucleation (kg kg$^{-1}$).

In WDM6 scheme, production rate for heterogeneous nucleation is calculated as the difference between $q_{I0}$ and the current ice mixing ratio ($q_I$):

$$P_{igen} = \frac{(4.92 \times 10^{-11} \frac{N_{ice}^{1.33}}{\rho} - q_I)}{\Delta t}$$

where $P_{igen}$ is the production rate of cloud-ice mass by heterogeneous nucleation (kg kg$^{-1}$ s$^{-1}$), and $\Delta$ t is the model time step (100 s).

We have revised the corresponding description in Lines 224–230 as follows:

WDM6 uses the formula $\rho q_{I0}(kg\ m^{-3}) = 4.92 \times 10^{-11} N_{ice}^{1.33}$ and $P_{igen}(kg kg^{-1} s^{-1}) = \frac{(q_{I0} - q_I)}{\Delta t}$ to calculate nucleation of ice from vapor due to the IN increase. Where, ρ denotes the air density, and $q_{I0}$ is the predicted ice mixing ratio after accounting for newly formed ice from heterogeneous nucleation (kg kg$^{-1}$). production rate for heterogeneous nucleation is calculated as the difference between $q_{I0}$ and the current ice mixing ratio ($q_I$). However, it does not account for the influence of nucleated IN size or the specific characteristics of different heterogeneous ice nucleation mechanisms on ice crystal development.

**21.    L212: Does "ρi" mean the density of ice? Why take the value of 500 kg/m3?**

A: Yes, ρ$_i$ mean the density of cloud ice. We used a constant value of 500 kg m$^{-3}$ following Park and Lim (2023). This value is also well-established in the literature and has been used in other research (Reisner et al., 1998; Morrison and Gettelman, 2008).

**22.  Equation (5): Please provide the reference for equation (5).**

A:Yes, We have revised the original writing into line 243-246:

Considering ice crystals generally grow from smaller particles and the radius of initial ice crystal size are often smaller than observed values, and with reference to the bin sizes of aerosol particles in CUACE (Um et al., 2018; Chen et al., 2021; Yang et al., 2021), this study assumes the ice crystal radius of $r_{df}^{\blacksquare}$ and $r_{if}^{\blacksquare}$ to be:

**23.  L252-255: It is difficult to understand this sentence.**

A:Thank you for pointing this out. We have revised the sentence to make our intention clearer. The purpose is to explain that, based on previous radar and modeling studies showing that dust mainly participates in cloud-ice processes between mid-tropospheric layer (-20 - 0 °C), the simulated dust distribution in this altitude range is used in our study to determine the dust-affected region.
We have revised the corresponding description in Lines 251–255 as follows:
Considering that many radar observations and model studies have indicated that dust mainly participates in within the mid-tropospheric layer (-20 - 0 ℃) between 4 and 7 km in altitude (Haarig et al., 2019; He et al., 2021; He et al., 2023), Fig. 1c also shows the simulated dust within this layer.

**24.  L264-265: The output interval of 3 h is too long to investigate the microphysical effects of IN. The cloud system might change evidently during this time period.**

A:We completely agree that the 3-hour output interval was too coarse to accurately resolve the microphysical processes influenced by IN. The original setting was primarily chosen to match the 6-hour cumulative precipitation data from the China Meteorological Administration used in our initial analysis.

Following your suggestion, we have rerun the model with a 1-hour output interval. To precisely align with the observed dust-precipitation events, we

interpolated the model results from the hour before and after to the exact observation time. All analyses in the revised manuscript pertaining to the temporal evolution of cloud microphysics now utilize this new, high-frequency (1-hour) dataset.

We have revised the corresponding description in Lines 264–265 as follows:

The model outputs 1-hourly precipitation data. To compare with the observed 6-hourly precipitation, the model outputs are temporally interpolated to the time stamps of the observations.

**25.** **L272: As I know, there are three types of horizontal resolution of NCEP FNL data, i.e., 2.5°, 1°, and 0.25°. Please provide the link for the NCEP FNL data with resolution f 0.15°.**

**A:** Yes, You are correct that the NCEP FNL dataset is available at spatial resolutions of 2.5 °, 1 °, and 0.25 °, rather than 0.15 °.

We have revised the corresponding description in Lines 301–303 as follows:

The initial and boundary meteorological conditions for GRAPES/CUACE are obtained from the NCEP/NCAR Final Operational Global Analysis (FNL) data, with a temporal resolution of 6 hours and a spatial resolution of 0.25 °.

For completeness, we have also added the official data access link in the revised manuscript:

The NCEP/NCAR Final Operational Global Analysis (FNL) data, with a temporal resolution of 6 hours and a spatial resolution of 0.25(https://rda.ucar.edu/datasets/ds083.3/).

**26.** **L293: The equation of aMAPE has been introduced in equation (8).**

**A:** Yes. We have revised the corresponding description in Lines 324–330 as follows:

The $aMAPE$ is used to evaluate whether the simulated precipitation is overestimated or underestimated compared with the observation. When aMAPE > 0, precipitation is overestimated; when aMAPE < 0, precipitation is underestimated.

27.    **L293-294: It is difficult to understand this sentence.**

A: Yes. We have revised the corresponding description in Lines 324–330 as follows:

The $aMAPE$ is used to evaluate whether the simulated precipitation is overestimated or underestimated compared with the observation. When aMAPE > 0, precipitation is overestimated; when aMAPE < 0, precipitation is underestimated.

28.    **L299: What is the vertical resolution of the simulations? I mean how many levels are included between 3 km and 5 km?**

**A:** In the GRAPES/CUACE model configuration used in this study, the atmosphere is divided into 32 vertical layers. Within the height range of 3–5 km, there are four model layers, located approximately at 3.11 km, 3.67 km, 4.25 km, and 4.86 km.

29.    **Figures 2a-c: Please introduce how to calculate the IN number. Is it the number of dust particles with diameter exceeding 0.5 μm?**

**A:** Thank you for the comment. The IN number in Figures 2a and 2b is not defined as the number of dust particles larger than 0.5 μm. Instead, it represents the activated ice-nucleating particle (IN) concentration, which is calculated in Equations (1) ,(2) and (3).
We have revised the original writing into line 330-338:

During the DP event, the implemented on-line aerosol–IN nucleation scheme enables dust aerosols to modify the nucleated IN number concentration. Figures 2a and 2b show the horizontal distribution of the maximum nucleated IN number concentration between 4 and 7 km above ground level at DP stations during the time period from 00:00 UTC on 11 April to 00:00 UTC on 15 April 2018 for T_CTL and T_IN, respectively. Figure 2c presents the vertical distribution of DP-event-averaged production rate for Nigen for T_CTL (red line) and T_IN (blue line). Figure 2d presents the vertical distribution of cloud ice mass production rate for heterogeneous ice nucleation for T_CTL and T_IN.

30.      Figure 2d: I don't understand the label of x-axis. Is it the IN nucleation rate? If so, the unit of nucleation rate should be #/kg/s (number of newly nucleated ice crystal per second) or g/kg/s (mass of newly nucleated ice crystal per second). Furthermore, the authors are encouraged to compare the nucleation rate of different types of regimes, such as deposition nucleation, condensation freezing, immersion freezing, and homogeneous freezing.

A: Yes, you are right. We have revised the figure to clarify that it represents the heterogeneous nucleation rate, and we have updated the units to g/kg/s, which is more appropriate.

And regarding the comparison among different freezing regimes, the immersion freezing process is indeed the dominant heterogeneous nucleation mechanism. Based on our simulations, the mass growth rate of deposition nucleation and condensation freezing together is only about4–5 orders of magnitude of that of immersion freezing. As for homogeneous freezing, it occurs essentially instantaneously and is not explicitly represented by a separate nucleation rate parameter.

We have added the following explanation into the line 363-366:

Moreover, immersion freezing is the dominant heterogeneous nucleation mechanism, exceeding deposition and condensation freezing by 4–5 orders of

magnitude in DP-event-averaged production rate for nucleated IN number concentration and 5–6 orders of magnitude in production rate of cloud ice.

**31.    Figures 2c-d: The figure title mentioned the results are from T_CCN and T_IN, but the legends in both panels show T_CTL and T_IN.**

A: Thank you for pointing out this inconsistency. We have corrected the figure titles and clarified the experiment definitions to ensure consistency throughout the manuscript.

In the revised manuscript, we use T_CTL and T_IN consistently.  Here, T_CTL represents the control experiment in which aerosols affect cloud condensation nuclei (CCN) only, which corresponds to the T_CCN experiment in the original version. T_IN represents the experiment in which aerosols affect both CCN and ice nuclei (IN) through the online aerosol–IN nucleation scheme, allowing us to isolate the impact of dust–IN interactions on cloud microphysics and precipitation.

**32.    I didn't find discussion or explanation about Figure 2 but only introduction of the figures at L298-305.**

A: Thank you for this comment. We agree that the original manuscript mainly described Figure 2 without sufficient physical interpretation. In the revised manuscript, we have added a dedicated discussion of Figure 2 in Section 3.1.

We have revised the original writing into line 330-338:

During the DP event, the implemented on-line aerosol–IN nucleation scheme enables dust aerosols to modify the nucleated IN number concentration. Figures 2a and 2b show the horizontal distribution of the maximum nucleated IN number concentration between 4 and 7 km above ground level at DP stations during the time period from 00:00 UTC on 11 April to 00:00 UTC on 15 April 2018 for T_CTL and T_IN, respectively. Figure 2c presents the vertical distribution of DP-event-averaged production rate for Nigen for T_CTL (red line) and T_IN (blue line). Figure 2d presents the vertical distribution of cloud ice mass production rate for heterogeneous ice nucleation for T_CTL and T_IN.

**33.** **Figures 3 and 4: What does "event averaged hydrometeors" mean in the figure title? And it should be "averaged mass concentration of different types of hydrometeors" instead of "averaged hydrometeor".**

A: Thank you for pointing this out. You are correct that the original wording was unclear. In this study, "event-averaged hydrometeors" refers to the time-averaged mass concentrations of different hydrometeor species during dust–precipitation (DP) events, rather than a single hydrometeor quantity.

Specifically, the averaging is performed over stations influenced by dust (defined by $PM_{2.5}/PM_{10} < 0.6$) and with precipitation amounts greater than 0.1 mm, focusing on the vertical distributions of hydrometeor mass concentrations during DP events.

To avoid ambiguity, we have revised the figure titles to "event-averaged mass concentrations of different types of hydrometeors", which more accurately reflects the content shown in Figures 3 and 4.

**34.** **Figure 4: What does Qv stand for? Is that water vapor mixing ratio? The water vapor is not a kind of hydrometeors, and the Qv profiles were not referred in the main text.**

A: Thank you for this comment. Qv denotes the water vapor mixing ratio. We agree with the reviewer that water vapor is not a hydrometeor and that the Qv profiles were not explicitly discussed in the original manuscript.

Since this study focuses on the impacts of dust aerosols on cloud hydrometeors and precipitation, the inclusion of Qv in Figure 4 was not essential. Therefore, we have removed the Qv profiles from Figure 4 and revised the figure accordingly. All references to Figure 4 have been updated to ensure consistency with the revised content.

**35.** **L337-339: What is the reason for higher temperature in T_IN case than the other 2 cases? Moreover, how does the temperature change of 0.1-0.5 ℃ lead to so remarkable**

reduction of ice crystal mixing ratio? The authors are suggested to explain this question in more detail. And if the warmer environment is the reason for the reduced mass of ice crystal, the IN nucleation rate should be decreased, rather than increased at 4-6 km compared between T_IN and T_CTL as shown in Figure 2d. Moreover, why did IN nucleation occur below 4 km with temperature above 0 ℃? It does not make sense.

A: Thank you for this insightful comment. We agree that the relationship between temperature changes, ice nucleation, and cloud-ice mass requires careful clarification.

After re-examining the thermodynamic and microphysical budgets, we find that although individual microphysical processes (e.g., deposition, riming, heterogeneous nucleation, and evaporation) contribute differently to heating or cooling tendencies, the net temperature differences between T_IN and the other experiments remain small (on the order of 0.1–0.5 ℃). Therefore, these temperature differences alone cannot directly explain the substantial changes in cloud-ice mass concentration.

Taking the 4–6 km layer as an example, warming-related microphysical processes include snow deposition (Psdep), graupel deposition (Pgdep), ice deposition (Pidep), accretion processes (Psacr, Pgacr, Piacr), rain accretion (Paacw), and heterogeneous ice nucleation (Pigen), while evaporative cooling is mainly associated with rain evaporation (Prevp).    Although the combined heating rates from these processes in T_IN are approximately 70%–95% of those in T_CTL, the resulting temperature changes remain about −0.002 ℃ due to cloud microphysics. This indicates that microphysical heating is not the dominant driver of the reduced cloud-ice mass.

Accordingly, we have removed the previous discussion that attributed cloud-ice reduction primarily to temperature changes and revised the manuscript to emphasize that the dominant mechanism is the suppression of depositional growth due to increased ice number concentration, rather than thermodynamic warming.

**36.** **L344: I can not find in Figure 4 at which level the mass concentration of ice crystal is reduced up to 0.1 g/kg. The maximum value of Qi is smaller than 0.05 g/kg as shown in Figure 3.**

A: Thank you for this insightful comment.

During the revision, we reprocessed the model output by increasing the temporal resolution from 3-hourly to 1-hourly output and reanalyzed the vertical distributions of hydrometeor mass concentrations, cloud-ice number concentrations, and ice nuclei concentrations. Based on the updated analysis, the maximum reduction in cloud-ice mass concentration occurs at approximately 8 km above ground level, with a decrease of about 0.025 g kg$^{-1}$, corresponding to roughly 15% of the cloud-ice mass concentration in T_CTL, rather than 0.1 g kg$^{-1}$.

We have corrected the relevant description in the manuscript to ensure consistency with Figures 3 and 4 and removed the inaccurate value reported previously.

**37.** **L345-347: Here the authors mentioned that "cloud ice increases" but in the last sentence they just wrote "cloud ice mixing ratio decreases by…". I am very confused about the inconsistency of the expressions.**

A: Thank you for pointing out this inconsistency. We agree that the original wording was unclear and potentially misleading because it did not clearly distinguish between cloud ice number concentration and cloud ice mass (mixing ratio). In the revised manuscript, we have clarified the expressions by explicitly specifying whether we refer to cloud ice number concentration or cloud ice mixing ratio in each case.

**38.** **L356-359: I don't understand what does cloud ice "transforms into cloud water" mean? Is it melting of cloud ice into liquid water? if so, the height for the increment of cloud water should be below 0 ℃ layer for melting occurs. But the difference in Qc peaks above 0 ℃ layer for phase 1 (Figure 4a, d). Please clarify it.**

**A:** During the revision, we reprocessed the model output by increasing the temporal resolution from 3-hourly to 1-hourly output and reanalyzed the vertical distributions of hydrometeor mass concentrations as well as cloud-ice and ice-nuclei number concentrations. Based on the updated analysis, we find that cloud water mixing ratio in T_IN is reduced to 90%-95% of that in T_CTL the 0–4 km layer, rather than increased. This reduction is mainly caused by by dust suppressing the production rate for cloud droplet activation from CCN in warm clouds (pcact), which decreases by about 5% in T_IN relative to T_CTL.

39.  L362-363: Please provide enough evident for competing available water vapor between INs and CCNs, such as comparing the diffusion growth rate of cloud droplet and ice crystal.

A: Thank you for this comment. We agree that clearer evidence is needed to support the competition for available water vapor between INs and CCNs.

After introducing the on-line aerosol–IN nucleation scheme, we re-examined the cloud microphysical budgets. The results show that the CCN-driven droplet activation and condensational growth in warm clouds are indeed weakened. Specifically, CCN-driven cloud droplet activation from CCN (Pcact) in T_IN decreases by about 5% compared to T_CTL, indicating that less water vapor is converted into cloud water. Cloud water mixing ratio in T_IN is reduced to 90%-95% of that in T_CTL the 0–4 km layer

40.  Figures 3 and 4 show the vertical profile of mixing ratio of graupel. But it seems not referred in the main text.

A: Thank you for the comment. We agree that graupel mixing ratio is shown in Figures 3 and 4 but was not explicitly discussed in the original text.

In our simulations, the graupel mixing ratio is substantially smaller than those of cloud ice and snow throughout the vertical column. During the dust–precipitation

event, graupel is mainly distributed around 5–6 km, and its change in T_IN relative to T_CTL is modest, remaining within approximately 90%–100% of T_CTL. Compared with the pronounced responses of cloud ice and snow, the contribution of graupel to the overall hydrometeor budget and precipitation response is therefore relatively minor.

For this reason, the discussion in the manuscript focuses on cloud ice, snow, cloud water, and rainwater, which exhibit much stronger sensitivity to the aerosol–IN nucleation scheme.

41. **Figure 6: It seems that the improvement of CCN or IN nucleation contributes insignificantly to the changes in precipitation pattern. The inherent defects of the numerical model (e.g., microphysics, dynamics, or the initial and boundary conditions) may play more important role in the evolution of cloud field and spatial distribution of precipitation.**

A: Thank you for this insightful comment. We agree that the improvement of CCN or IN nucleation alone does not lead to a dramatic change in the large-scale precipitation pattern, and that uncertainties associated with model dynamics, microphysics, and initial and boundary conditions can play an important role in shaping the spatial distribution of precipitation.

From a cloud-microphysical perspective, this limited precipitation response is physically consistent with our results. Although the on-line aerosol–IN nucleation scheme modifies cloud microphysical processes, the magnitude of these changes remains relatively small in the lower troposphere, where precipitation forms. Specifically, in T_IN, the cloud water and rainwater mixing ratios below 4 km are reduced by only about 5–10% compared to T_CTL. Such modest reductions in liquid-phase hydrometeors lead to correspondingly small changes in surface precipitation.

This behavior is consistent with previous modeling studies (e.g., Park and Lim, 2023), which also reported that dust only had a weak influence on precipitation amount and pattern. Therefore, our results suggest that dust aerosols primarily

modulate cloud microphysical structures rather than acting as a dominant control on precipitation distribution during this event.

We have clarified this point in the revised manuscript to better distinguish between microphysical impacts and precipitation-scale responses in line 506-512:

In summary, because the reduction in cloud water in the 0–4 km layer is relatively small, the corresponding decrease in rainwater reaching the surface is also limited. As a result, the on-line aerosol–IN nucleation scheme exerts only a weak influence on the total precipitation amount. Nevertheless, it can modulate the spatial and temporal distribution of precipitation, impressing overestimated and altering underestimation in a degree, which is consistent with the findings of Park and Lim (2023) and Su and Fung (2018b).

**42.    L403-404: From which figure the authors found "the suppressed cloud water is transported downstream in T_IN"?**

**A:** Thank you for this question. The statement regarding downstream transport of suppressed cloud water is not inferred directly from Figures 3 or 4, but from an additional diagnostic analysis of hydrometeor fluxes in line 482-497:

We calculate horizontal hydrometeor fluxes across 116 °E, 33 °–50 °N and 33 °N, 103 °–116 °E from 12:00 UTC on 12 April to 18:00 UTC on 13 April (Fig. 6). Over the entire 0–12 km layer, the total hydrometeor flux slightly increases to about 102% of that in T_CTL.

Within the temperature range from 0 to -40 ℃, the total horizontal hydrometeor flux decreases by about 11 %, primarily due to a substantial reduction in cloud ice flux, accompanied by increases in snow and graupel fluxes. In Layer A, the total hydrometeor flux is about $4.4 \times 10^{-5}$ kg s$^{-1}$, corresponding to about 75 % of T_CTL. Cloud ice flux drops sharply to about 8 % of T_CTL, while snow and graupel fluxes increase markedly to about 19.8 times and 7.8 times, respectively. In Layer B, the

total hydrometeor flux is about $2.6 \times 10^{-6}$ kg s$^{-1}$, corresponding to about 93 % of T_CTL, with cloud ice flux reduced to about 28 % of T_CTL, and snow and graupel fluxes increased to about 2.3 times and about 1.8 times, respectively. At temperatures above 0 ℃, the total horizontal hydrometeor flux increases to about 106 % of T_CTL, with cloud water and rainwater fluxes increasing to about 115 % and about 108 %, respectively.

**43.    L419-422: The authors are suggested to explain this results in more detail.**

A: Thank you for this comment. Under the influence of dust, in 0–4 km, the production rate for cloud droplet activation from CCN (PCACT) in T_IN decreases by about 5% relative to T_CTL, indicating that less water vapor is converted into cloud water.    As a result, the cloud water mixing ratio in T_IN is reduced to approximately 90%–95% of that in T_CTL within the 0–4 km layer.

Because the reduction in cloud water is relatively small, the corresponding decrease in rainwater reaching the surface is also limited, leading to only a weak response of surface precipitation to dust perturbations. This behavior is consistent with previous studies (Park and Lim, 2023; Su and Fung, 2018b).

In summary, because the reduction in cloud water in the 0–4 km layer is relatively small, the corresponding decrease in rainwater reaching the surface is also limited. As a result, the on-line aerosol–IN nucleation scheme exerts only a weak influence on the total precipitation amount. Nevertheless, it can modulate the spatial and temporal distribution of precipitation, which is consistent with the findings of Park and Lim (2023) and Su and Fung, (2018).

**44.    L436: I can't find from which figure the authors reached the conclusion that "IN concentrations reached 103-104 $L$-1 between 3 and 5 km altitude". Figure 2c shows the maximum number concentration is between 102 and 103 $L^{-1}$.**

A: Thank you for pointing this out. The confusion arises from an unclear distinction between maximum and event-averaged IN number concentrations in the original manuscript.

Figures 2a and 2b show the maximum nucleated IN number concentrations between 3 and 5 km during the dust–precipitation event, whereas Figure 2c presents the event-averaged vertical profiles of IN concentrations over all DPA stations. As a result, the peak values shown in T_IN can reach $10^{3}$–$10^{4}$ L$^{-1}$, while the peak values shown in T_IN can reach $10^{0}$–$10^{1}$ L$^{-1}$.

**45.      L430-438: I did not find the discussion related to the number concentration of IN in Section 3 Results.**

A: Thank you for pointing this out.    We agree that the discussion of IN number concentration was not sufficiently explicit in the original version of Section 3.1.

In the revised manuscript, we have clarified and strengthened the discussion of IN number concentration by explicitly describing its vertical distribution, magnitude, and differences among experiments (T_CTL, and T_IN), particularly in relation to Figure 2.

**46.      L440-441: Please explain why "dust suppresses the formation of ice-phase hydrometeors"?**

A: Thank you for this important question. The mechanisms by which dust suppresses ice-phase hydrometeor formation differ across vertical layers, mainly due to distinct thermodynamic conditions and dominant microphysical processes.

Above 7 km (temperatures below $-17$ ℃), the introduction of the online aerosol–IN nucleation scheme leads to a decrease in IN number concentration in T_IN compared to T_CTL (Fig. 2c). As a result, cloud-ice number concentrations in

T_IN are approximately 5 L$^{-1}$ lower than in T_CTL, corresponding to about 70% of T_CTL (Fig. 3d), while the cloud-ice mass concentration is reduced to only 10%–50% of T_CTL (Fig. 3a,b). This reduction is primarily caused by a strong suppression of the total cloud-ice formation processes—heterogeneous ice nucleation (Pigen) and vapor deposition growth of cloud ice (Pidep)— in this layer, which decreases to less than 24% of that in T_CTL. On the one hand, the reduced IN number concentration directly weakens Pigen by 1–2 orders of magnitude relative to T_CTL. On the other hand, the lower cloud-ice number concentration allows ice crystals to grow to larger sizes, with effective diameters reaching 98%–135% of those in T_CTL. This shift toward fewer but larger ice crystals reduces the total surface area available for vapor deposition, thereby limiting the overall efficiency of Pidep. Consequently, Pidep decreases to 20%–50% of T_CTL, with the strongest suppression occurring near 7–8 km.

Between 4 and 7 km (temperatures approximately $-17$ ℃ to $-2$ ℃), the enhanced activation of ice nuclei in T_IN leads to an increase in cloud-ice number concentration through stronger heterogeneous ice nucleation. However, the resulting increase in ice crystal number causes a pronounced decrease in effective diameters of cloud ice which decreases to only 77%–97% of T_CTL. The smaller ice crystals grow less efficiently by vapor deposition, substantially suppressing Pidep and limiting the accumulation of ice-phase mass despite the higher ice crystal number concentration.

In summary, dust aerosols suppress the formation of ice-phase hydrometeors through different mechanisms at different altitudes: by reducing both ice nucleation and deposition growth in the upper troposphere, and by enhancing ice number concentration but inhibiting depositional growth efficiency in the mid-troposphere. These combined effects ultimately lead to a net reduction in ice-phase hydrometeor mass.

**47.** **L442-444: I can't understand this sentence. First, why does "higher cloud-top temperature" and "more small-sized ice-phase cloud particles"? The warm environment should suppress the IN nucleation. Second, why "both of which could limit ice-phase hydrometeor development"? Does "ice-phase hydrometeor" include "ice-phase cloud particles"? It makes me confused.**

A: Thank you for pointing out this ambiguity. We agree that the original sentence was unclear and could lead to misunderstanding. We have revised the text to clarify both the physical meaning and terminology. The main points are explained as follows.

First, the term "higher cloud-top temperature" does not imply that a warmer environment directly enhances ice nucleation. Instead, it reflects a secondary thermodynamic response to changes in cloud microphysical processes. As discussed in question 34, the differences in temperature among the experiments are small (on the order of 0.1–0.5 ℃) and do not control the ice-nucleation rate. The enhanced ice nucleation at 4–7 km in T_IN is driven by increased availability of ice-nucleating particles, rather than by temperature changes.

Second, the phrase "more small-sized ice-phase cloud particles" refers to the microphysical consequence of enhanced heterogeneous ice nucleation. The increase in activated IN between 4 and 7 km (temperature approximately −17 °C to -2 ℃) leads to a larger number of ice crystals, but with reduced effective diameters of cloud ice to only 77%–97% of T_CTL. This size reduction suppresses depositional growth efficiency and limits the accumulation of ice-phase mass, even though ice crystal number concentration increases.

Third, in this study, the term "ice-phase hydrometeors" refers to the sum of the cloud ice and snow. To avoid confusion, we have rewritten the sentence by removing the misleading reference to cloud-top temperature and by explicitly distinguishing between ice crystal number concentration and ice-phase mass. The revised text now emphasizes that dust affects ice-phase clouds mainly through microphysical pathways

associated with ice crystal size and depositional growth, rather than through direct temperature effects.

**48.    L453-456: Please explain how "increasing production rate for nucleation of ice suppress the precipitation". Moreover, what is the relationship between suppressed precipitation and reduced cloud and rain water? Please state it in more detail.**

**A:** Thank you for this comment.

Cloud water and rainwater are mainly distributed in layer c (temperature approximately -2 ℃ to 18 ℃). In this layer, both cloud-water and rainwater mixing ratios in T_IN are 90%-95% of those in T_CTL by approximately. This reduction is primarily attributed to a weakening of the production rate for cloud droplet activation from CCN (Pcact), which decreases by about 5% in T_IN relative to T_CTL, indicating a suppressed conversion of water vapor into liquid water. As a consequence of the reduced cloud-water content, the production rate for accretion of cloud rain by cloud water (Pracw) are also weakened, with reductions of approximately 5%–10%. Meanwhile, the conversion of rainwater into ice-phase hydrometeors (Psaci, Pgaci, and Piaci) is enhanced. However, under the thermodynamic conditions of layer c, temperatures exceed the melting thresholds of ice-phase hydrometeors, and newly formed snow and graupel rapidly melt and are converted back into rainwater. Consequently, dusts lead to a limited reduction in surface precipitation.

**Related References**

Cantrell, W., Bunker, K., Niehaus, J., China, S., Woodward, X., Kostinski, A., and Mazzoleni, C.: Ice nucleation in the contact mode: Temperature and size dependence for selected dusts, AIP Conference Proceedings, 1527, 926, https://doi.org/10.1063/1.4803423, 2013.

Chen, J., Wu, Z., Chen, J., Reicher, N., Fang, X., Rudich, Y., and Hu, M.: Size-resolved atmospheric ice-nucleating particles during East Asian dust events,

Atmospheric Chemistry and Physics, 21, 3491–3506, https://doi.org/10.5194/acp-21-3491-2021, 2021.

Chen, Q., Yin, Y., Jiang, H., Chu, Z., Xue, L., Shi, R., Zhang, X., and Chen, J.: The Roles of Mineral Dust as Cloud Condensation Nuclei and Ice Nuclei During the Evolution of a Hail Storm, Journal of Geophysical Research Atmospheres, 124, https://doi.org/10.1029/2019JD031403, 2019.

DeMott, P. J., Prenni, A. J., McMeeking, G. R., Sullivan, R. C., Petters, M. D., Tobo, Y., Niemand, M., Möhler, O., Snider, J. R., Wang, Z., and Kreidenweis, S. M.: Integrating laboratory and field data to quantify the immersion freezing ice nucleation activity of mineral dust particles, Atmospheric Chemistry and Physics, 15, 393–409, https://doi.org/10.5194/acp-15-393-2015, 2015.

DeMott, P. J., Hill, T. C. J., Petters, M. D., Bertram, A. K., Tobo, Y., Mason, R. H., Suski, K. J., McCluskey, C. S., Levin, E. J. T., Schill, G. P., Boose, Y., Rauker, A. M., Miller, A. J., Zaragoza, J., Rocci, K., Rothfuss, N. E., Taylor, H. P., Hader, J. D., Chou, C., Huffman, J. A., Pöschl, U., Prenni, A. J., and Kreidenweis, S. M.: Comparative measurements of ambient atmospheric concentrations of ice nucleating particles using multiple immersion freezing methods and a continuous flow diffusion chamber, Atmospheric Chemistry and Physics, 17, 11227–11245, https://doi.org/10.5194/acp-17-11227-2017, 2017.

Eastwood, M. L., Cremel, S., Gehrke, C., Girard, E., and Bertram, A. K.: Ice nucleation on mineral dust particles: Onset conditions, nucleation rates and contact angles, Journal of Geophysical Research: Atmospheres, 113, https://doi.org/10.1029/2008JD010639, 2008.

Haarig, M., Ansmann, A., Walser, A., Baars, H., Urbanneck, C., Weinzierl, B., Schöberl, M., Dollner, M., Mamouri, R., and Althausen, D.: Estimation of dust related ice nucleating particles in the atmosphere: Comparison of profiling and

in-situ measurements, E3S Web Conf., 99, 04002, https://doi.org/10.1051/e3sconf/20199904002, 2019.

Han, Y., CHEN, Y., Fang, X., and Zhao, T.: The possible effect of dust aerosol on precipitation in Tarim basin., China Environmental Science, 2008, 5 ( 102-106 ), https://doi.org/10.3321/j.issn:1000-6923.2008.02.002, 2008.

He, C., Yin, Y., Huang, Y., Kuang, X., Cui, Y., Chen, K., Jiang, H., Kiselev, A., Möhler, O., and Schrod, J.: The Vertical Distribution of Ice-Nucleating Particles over the North China Plain: A Case of Cold Front Passage, Remote Sensing, 15, 4989, https://doi.org/10.3390/rs15204989, 2023.

He, Y., Zhang, Y., Liu, F., Yin, Z., Yi, Y., Zhan, Y., and Yi, F.: Retrievals of dust-related particle mass and ice-nucleating particle concentration profiles with ground-based polarization lidar and sun photometer over a megacity in central China, Atmospheric Measurement Techniques, 14, 5939–5954, https://doi.org/10.5194/amt-14-5939-2021, 2021.

Hiranuma, N., Augustin-Bauditz, S., Bingemer, H., Budke, C., Curtius, J., Danielczok, A., Diehl, K., Dreischmeier, K., Ebert, M., Frank, F., Hoffmann, N., Kandler, K., Kiselev, A., Koop, T., Leisner, T., Möhler, O., Nillius, B., Peckhaus, A., Rose, D., Weinbruch, S., Wex, H., Boose, Y., DeMott, P. J., Hader, J. D., Hill, T. C. J., Kanji, Z. A., Kulkarni, G., Levin, E. J. T., McCluskey, C. S., Murakami, M., Murray, B. J., Niedermeier, D., Petters, M. D., O'Sullivan, D., Saito, A., Schill, G. P., Tajiri, T., Tolbert, M. A., Welti, A., Whale, T. F., Wright, T. P., and Yamashita, K.: A comprehensive laboratory study on the immersion freezing behavior of illite NX particles: a comparison of 17 ice nucleation measurement techniques, Atmospheric Chemistry and Physics, 15, 2489–2518, https://doi.org/10.5194/acp-15-2489-2015, 2015.

Hu, Y., Tian, P., Huang, M., Bi, K., Schneider, J., Umo, N. S., Ullmerich, N., Höhler, K., Jing, X., Xue, H., Ding, D., Liu, Y., Leisner, T., and Möhler, O.:

Characteristics of ice-nucleating particles in Beijing during spring: A comparison study of measurements between the suburban and a nearby mountain area, Atmospheric Environment, 293, 119451, https://doi.org/10.1016/j.atmosenv.2022.119451, 2023.

Ilotoviz, E., Khain, A. P., Benmoshe, N., Phillips, V. T. J., and Ryzhkov, A. V.: Effect of Aerosols on Freezing Drops, Hail, and Precipitation in a Midlatitude Storm, Journal of the Atmospheric Sciences, 73, 109–144, https://doi.org/10.1175/JAS-D-14-0155.1, 2016.

Jiang, H., Yin, Y., Wang, X., Gao, R., Yuan, L., Chen, K., and Shan, Y.: The measurement and parameterization of ice nucleating particles in different backgrounds of China, Atmospheric Research, 181, 72–80, https://doi.org/10.1016/j.atmosres.2016.06.013, 2016.

Kaufman, Y. J., Tanré, D., and Boucher, O.: A satellite view of aerosols in the climate system, Nature, 419, 215–223, https://doi.org/10.1038/nature01091, 2002.

Lee, S. S., Kim, B.-G., Yum, S. S., Seo, K.-H., Jung, C.-H., Um, J. S., Li, Z., Hong, J., Chang, K.-H., and Jeong, J.-Y.: Effects of aerosol on evaporation, freezing and precipitation in a multiple cloud system, Clim Dyn, 48, 1069–1087, https://doi.org/10.1007/s00382-016-3128-1, 2017.

Liu, H., Yu, Y., Xia, D., Zhao, S., Ma, X., and Dong, L.: Analysis of the relationship between dust aerosol and precipitation in spring over East Asia using EOF and SVD methods, Science of The Total Environment, 908, 168437, https://doi.org/10.1016/j.scitotenv.2023.168437, 2024.

Morrison, H. and Gettelman, A.: A New Two-Moment Bulk Stratiform Cloud Microphysics Scheme in the Community Atmosphere Model, Version 3 (CAM3). Part I: Description and Numerical Tests, https://doi.org/10.1175/2008JCLI2105.1, 2008.

Naeger, A. R.: Impact of dust aerosols on precipitation associated with atmospheric rivers using WRF-Chem simulations, Results in Physics, 10, 217–221, https://doi.org/10.1016/j.rinp.2018.05.027, 2018.

Niehaus, J., Becker, J. G., Kostinski, A., and Cantrell, W.: Laboratory Measurements of Contact Freezing by Dust and Bacteria at Temperatures of Mixed-Phase Clouds, https://doi.org/10.1175/JAS-D-14-0022.1, 2014.

Pan, X., Uno, I., Wang, Z., Nishizawa, T., Sugimoto, N., Yamamoto, S., Kobayashi, H., Sun, Y., Fu, P., Tang, X., and Wang, Z.: Real-time observational evidence of changing Asian dust morphology with the mixing of heavy anthropogenic pollution, Sci Rep, 7, 335, https://doi.org/10.1038/s41598-017-00444-w, 2017.

Park, S.-Y. and Lim, K.-S. S.: Implementation of Prognostic Cloud Ice Number Concentrations for the Weather Research and Forecasting (WRF) Double-Moment 6-Class (WDM6) Microphysics Scheme, Journal of Advances in Modeling Earth Systems, 15, e2022MS003009, https://doi.org/10.1029/2022MS003009, 2023.

Patnaude, R. J., McCluskey, C. S., Roberts, G. C., DeMott, P. J., Hill, T. C. J., McFarquhar, G. M., Kollias, P., Ranjbar, K., Wolde, M., and Kreidenweis, S. M.: Characteristics of Ice Nucleating Particles From the Long-Range Transport of Saharan Dust, Geophysical Research Letters, 52, e2024GL113365, https://doi.org/10.1029/2024GL113365, 2025.

Possner, A., Ekman, A. M. L., and Lohmann, U.: Cloud response and feedback processes in stratiform mixed-phase clouds perturbed by ship exhaust, Geophysical Research Letters, 44, 1964–1972, https://doi.org/10.1002/2016GL071358, 2017.

Reisner, J., Rasmussen, R. M., and Bruintjes, R. T.: Explicit forecasting of supercooled liquid water in winter storms using the MM5 mesoscale model,

Quarterly Journal of the Royal Meteorological Society, 124, 1071–1107, https://doi.org/10.1002/qj.49712454804, 1998.

Rosenfeld, D. and Bell, T. L.: Why do tornados and hailstorms rest on weekends?, Journal of Geophysical Research: Atmospheres, 116, https://doi.org/10.1029/2011JD016214, 2011.

Stier, P., van den Heever, S. C., Christensen, M. W., Gryspeerdt, E., Dagan, G., Saleeby, S. M., Bollasina, M., Donner, L., Emanuel, K., Ekman, A. M. L., Feingold, G., Field, P., Forster, P., Haywood, J., Kahn, R., Koren, I., Kummerow, C., L'Ecuyer, T., Lohmann, U., Ming, Y., Myhre, G., Quaas, J., Rosenfeld, D., Samset, B., Seifert, A., Stephens, G., and Tao, W.-K.: Multifaceted aerosol effects on precipitation, Nat. Geosci., 17, 719–732, https://doi.org/10.1038/s41561-024-01482-6, 2024.

Su, L. and Fung, J. C. H.: Investigating the role of dust in ice nucleation within clouds and further effects on the regional weather system over East Asia – Part 2: modification of the weather system, Atmospheric Chemistry and Physics, 18, 11529–11545, https://doi.org/10.5194/acp-18-11529-2018, 2018.

Tobo, Y., Adachi, K., DeMott, P. J., Hill, T. C. J., Hamilton, D. S., Mahowald, N. M., Nagatsuka, N., Ohata, S., Uetake, J., Kondo, Y., and Koike, M.: Glacially sourced dust as a potentially significant source of ice nucleating particles, Nat. Geosci., 12, 253–258, https://doi.org/10.1038/s41561-019-0314-x, 2019.

Um, J., McFarquhar, G. M., Stith, J. L., Jung, C. H., Lee, S. S., Lee, J. Y., Shin, Y., Lee, Y. G., Yang, Y. I., Yum, S. S., Kim, B.-G., Cha, J. W., and Ko, A.-R.: Microphysical characteristics of frozen droplet aggregates from deep convective clouds, Atmospheric Chemistry and Physics, 18, 16915–16930, https://doi.org/10.5194/acp-18-16915-2018, 2018.

Wang, W.: Observation and study on the transport of dust aerosol and its climate effect., Doctoral dissertation, Lanzhou University, 2013.

Yang, J., Hu, X., Lei, H., Duan, Y., Lv, F., and Zhao, L.: Airborne Observations of Microphysical Characteristics of Stratiform Cloud Over Eastern Side of Taihang Mountains, Chinese Journal of Atmospheric Sciences, 45(1), 88−106, 2021.

Zhang, Y., Yu, F., Luo, G., Fan, J., and Liu, S.: Impacts of long-range-transported mineral dust on summertime convective cloud and precipitation: a case study over the Taiwan region, Atmospheric Chemistry and Physics, 21, 17433–17451, https://doi.org/10.5194/acp-21-17433-2021, 2021.

Zhou, C., Zhang, X., Gong, S., Wang, Y., and Xue, M.: Improving aerosol interaction with clouds and precipitation in a regional chemical weather modeling system, Atmospheric Chemistry and Physics, 16, 145–160, https://doi.org/10.5194/acp-16-145-2016, 2016.

---

## Author Comment (AC4)

**This study investigates how dust aerosols influence precipitation in China using an improved online aerosol–ice-nucleation (aerosol-IN) scheme implemented in the GRAPES/CUACE regional model. The topic is interesting and important in the field of aerosol–cloud–precipitation interactions. However, in many parts of the manuscript, the authors draw conclusions without sufficient observational evidence. This is the major drawback of the study. Therefore, I am on the negative side regarding publication of this paper.**

Dear reviewers,

Thank you for your thorough review of the manuscript. We read the reviewer's comments carefully, and have responded and taken your comments into consideration and revised the manuscript accordingly. All the changes have been highlighted in the revised manuscript. Our detailed responses, including a point-by-point response to the reviews and a list of all relevant changes, are as follows:

1. **While the study focuses on the mechanisms of dust's impact on ice nuclei, the abstract provides limited discussion on this aspect. Relevant conclusions should be supplemented.**

A: Thank you for this valuable comment. In the revised manuscript, we have supplemented the abstract to explicitly state how dust modifies ice nucleation processes by reducing cloud ice nucleation efficiency, altering the heterogeneous nucleation and deposition growth of cloud ice, and subsequently influencing precipitation development. These additions aim to better highlight the mechanistic focus of this study while remaining consistent with the detailed analyses presented in the main text.

In line 16 -41:

To investigate the impact of ice nuclei (IN) activated by dust aerosols on precipitation over China, this study uses regional Global/Regional Assimilation and Prediction System – China Meteorological Administration Unified Atmospheric Chemistry Environment (GRAPES/CUACE). The original temperature-dependent IN

nucleation scheme is improved by incorporating an on-line aerosol–IN nucleation one. The INs are then fed on-line into the Double-Moment 6-Class (WDM6) cloud microphysics scheme to study in a typical dust affected precipitation event in East Asia.

Dust modifies the spatial distribution and density of IN, impacting heterogeneous nucleation. Compared with the systematic underestimation in original WDM6, the peak values of nucleated INs can reach 10-4 L-1 with the improved scheme, which is closer to observations. Cloud ice is reasonably formed between 4 and 7 km in height.

Dust can inhibit the development of clouds. Above 7 km, dust suppresses the growth of cloud ice (both heterogeneous nucleation and deposition growth), and the total production rate of cloud ice drops to less than 24% of that in the control test T_CTL, promoting snow formation and ultimately reducing the total ice-phase hydrometeor content to 70–85% of T_CTL. Between 4 and 7 km, dust enhances heterogeneous nucleation of cloud ice, but suppresses cloud ice deposition growth, resulting in the total ice-phase hydrometeor content decreasing to 85–91% of T_CTL. Below 4 km, dust suppresses the conversion of water vapor to cloud water and of cloud water to rain, reducing the liquid-phase hydrometeor content to 90–95% of T_CTL.

Dust also modulates the precipitation distribution closer to observations. It suppresses precipitation near dust source areas, where mean precipitation decreased by about 4.5 mm, while the downstream event-mean precipitation increased by about 1.1 mm.

2. **The manuscript contains several formatting issues that require careful revision. For example, the font in "2.2 WDM6 microphysics scheme" is noticeably inconsistent.**

A: Thank you for pointing out these formatting issues. We have carefully checked the entire manuscript and corrected the inconsistent fonts, formatting errors, and typographical issues, including those in Section 2.2 ("WDM6 microphysics scheme").

3. **In Section 3, the analysis should focus more on phenomena and mechanisms. Descriptions of figures and tables, such as those in lines 298-304, can be directly included in the captions.**

A: Thank you for this valuable suggestion. Following this comment, and in combination with suggestions from other reviewers, we have substantially revised Section 3 to strengthen the discussion of physical phenomena and underlying mechanisms.

In the revised Section 3, the analysis now explicitly links the evolution of activated ice-nucleating particle concentrations to changes in cloud hydrometeor budgets, and further examines how dust aerosols regulate the transformation and redistribution of hydrometeors within the cloud system. In particular, we analyze how dust-induced suppression of cloud ice formation and the concurrent enhancement of snow and graupel production modify the hydrometeor composition and transport pathways. Please refer to the revised Section 3 for details.

4. **In lines 430-440, the improved on-line model simulates significantly higher ice crystal concentrations compared to the WDM6 results. What causes this? This is a key highlight of the study, yet the authors did not provide an in-depth or systematic explanation in Section 3. The authors should systematically analyze the mechanisms behind the improved simulation performance after the model enhancement.**

A: Thank you for pointing out this important issue.

In the revised manuscript, we have clarified the distinction among activated INP concentration, cloud ice number concentration, and ice-phase mass concentration, and systematically analyzed their relationships using combined diagnostics of INP activation, cloud ice number, hydrometeor mass, and microphysical budget terms. Our results show that:

Above 7 km, dust suppresses the growth of cloud ice (both heterogeneous nucleation and deposition growth), and the total production rate of cloud ice drops to less than 24% of that in the control test T_CTL, promoting snow formation and ultimately reducing the total ice-phase hydrometeor content to 70–85% of T_CTL. Between 4 and 7 km, dust enhances heterogeneous nucleation of cloud ice, but suppresses cloud ice deposition growth, resulting in the total ice-phase hydrometeor

content decreasing to 85–91% of T_CTL. Below 4 km, dust suppresses the conversion of water vapor to cloud water and of cloud water to rain, reducing the liquid-phase hydrometeor content to 90–95% of T_CTL.

A systematic explanation of these mechanisms has now been added to Section 3, with explicit references to the relevant figures and microphysical process rates.

5. **Finally, this study focuses on the impact of dust processes on ice nuclei and precipitation, utilizing model simulations. However, the analysis in Sections 3 and 4 provides limited discussion on the influence of the dust event, which is only described in "2.4 Case description and test design." The analysis of model results should incorporate the evolution of the dust process, rather than merely analyzing the simulated ice crystal and precipitation outcomes. In summary, this is a highly meaningful study, and the authors are encouraged to strengthen the analysis of the model results.**

A: Thank you for this constructive suggestion. We agree that the dust event itself should play a more explicit role in interpreting the simulated cloud and precipitation responses.

In the revised manuscript, we have strengthened the linkage between the dust process and the microphysical and precipitation responses in Sections 3 and 4 in the following ways.

In section 3.1 and 3.2, We reorganized the discussion of cloud microphysical processes to emphasize their dependence on dust-induced IN perturbations, particularly in the mid- and upper-tropospheric layers where dust influence is strongest. The diagnosed changes in cloud ice, snow, and associated budget terms are now interpreted in the context of the evolving dust plume rather than as isolated microphysical outcomes.

In section 3.3, the discussion of precipitation responses has been revised to highlight how dust-driven modifications in hydrometeor mass budget and hydrometeor flux and during the dust–precipitation period contributes to the spatial redistribution of precipitation, rather than changes in total precipitation alone.